# Maf-family bZIP transcription factor NRL interacts with RNA-binding proteins and R-loops in retinal photoreceptors

Ximena Corso Diaz[1,2]*, Xulong Liang[1], Kiam Preston[1], Bilguun Tegshee[1], Milton A English[1], Jacob Nellissery[1], Sharda Prasad Yadav[1], Claire Marchal[1,3], Anand Swaroop[1]*

[1]Neurobiology, Neurodegeneration and Repair Laboratory, National Eye Institute, National Institutes of Health, Bethesda, United States; [2]Department of Ophthalmology, Byers Eye Institute, Stanford University, Stanford, United States; [3]In silichrom Ltd, Newbury, United Kingdom

## eLife Assessment

This **important** study employed multiple orthogonal techniques and tissue samples to investigate the interaction between the NRL transcription factor and RNA-binding proteins in the retina. The findings are **convincing** to support an interaction between NRL and the DHX9 helicase. The significance of the study could be enhanced with functional experiments of NRL-R-loop interactions in the developing retina and their potential role in photoreceptor health and gene regulation.

**Abstract** RNA-binding proteins (RBPs) perform diverse functions including the regulation of chromatin dynamics and the coupling of transcription with RNA processing. However, our understanding of their actions in mammalian neurons remains limited. Using affinity purification, yeast-two-hybrid and proximity ligation assays, we identified interactions of multiple RBPs with neural retina leucine (NRL) zipper, a Maf-family transcription factor critical for retinal rod photoreceptor development and function. In addition to splicing, many NRL-interacting RBPs are associated with R-loops, which form during transcription and increase during photoreceptor maturation. Focusing on DHX9 RNA helicase, we demonstrate that its expression is modulated by NRL and that the NRL–DHX9 interaction is positively influenced by R-loops. ssDRIP-Seq analysis reveals both stranded and unstranded R-loops at distinct genomic elements, characterized by active and inactive epigenetic signatures and enriched at neuronal genes. NRL binds to both types of R-loops, suggesting an epigenetically independent function. Our findings suggest additional functions of NRL during transcription and highlight complex interactions among transcription factors, RBPs, and R-loops in regulating photoreceptor gene expression in the mammalian retina.

## Introduction

Cell type-specific and quantitatively precise decoding of genetic information produces a plethora of divergent cellular morphologies and functions during development (*Davidson and Levine, 2008*; *Spitz and Furlong, 2012*; *El-Danaf et al., 2023*). The initiation of DNA-dependent RNA synthesis by RNA Polymerase II is coordinated by complex interactions among chromatin modifying proteins, ubiquitous and cell type-specific transcription factors, and transcriptional machineries (*Cramer, 2019*; *Guo et al., 2019*; *Sharp et al., 2022*). Combinatorial and sequence-specific binding of trans-acting regulatory proteins to *cis*-regulatory elements including enhancers and promoters is facilitated by specific

*For correspondence:
ximenac@stanford.edu (XCD);
swaroopa@nei.nih.gov (AS)

genome topologies, resulting in productive transcriptional output by assembling phase-separated condensates (*Cramer, 2019*; *Sharp et al., 2022*). Concurrent control mechanisms meticulously guide every step of the genetic program. Curiously, cell type- and tissue-specific transcription factors are also reported to contribute to additional transcriptional and post-transcriptional regulatory steps, including 3D chromatin dynamics and splicing (*Han et al., 2017*; *Stadhouders et al., 2018*; *Kim and Shendure, 2019*), to achieve stringent spatiotemporal control of gene expression patterns.

Gene regulatory networks underlying neuronal differentiation in the mammalian retina (*Agathocleous and Harris, 2009*; *Cepko, 2014*; *Norrie et al., 2019*; *Lyu et al., 2021*; *Diacou et al., 2022*), especially photoreceptors (*Swaroop et al., 2010*; *Brzezinski and Reh, 2015*; *Cepko, 2015*), offer an attractive system to dissect divergent roles of transcription factors in controlling quantitatively precise cell type-specific gene expression. The Maf-family basic motif leucine zipper (bZIP) transcription factor neural retina leucine (NRL) determines rod photoreceptor cell fate as its loss leads to retina with only cone photoreceptors (*Mears et al., 2001*), whereas ectopic expression of NRL in developing cones results in generation of rods (*Oh et al., 2007*). Continued expression of NRL is essential for maintaining rod function (*Yu et al., 2017*), and mutations affecting NRL activity are associated with retinopathies (*Bessant et al., 1999*; *Nishiguchi et al., 2004*; *Kanda et al., 2007*; *El-Asrag et al., 2022*). NRL establishes rod cell identity and function by regulating precise transcriptional output of thousands of genes (*Hao et al., 2012*; *Kim et al., 2016*; *Liang et al., 2022*) in collaboration with cone rod homeobox (CRX) (*Mitton et al., 2000*; *Hennig et al., 2008*; *Corbo et al., 2010*) and other regulatory proteins (*Swaroop et al., 2010*; *Brzezinski and Reh, 2015*; *Kim et al., 2012*; *White et al., 2016*). Given its fundamental role as a primary activator of rod expressed genes and continuous high-level expression during the life of rod photoreceptors, we hypothesized the involvement of NRL in modulating additional steps during the transcription process. This prediction was strengthened by transcriptional activation synergy observed in rhodopsin regulation between NRL and NonO/p54$^{nrb}$ (*Yadav et al., 2014*). NonO, a ubiquitous nuclear paraspeckle protein, binds to DNA, RNA as well as proteins, and regulates distinct cellular events including the coupling of the circadian clock to cell cycle and transcription to splicing (*Fox and Lamond, 2010*; *Kowalska et al., 2013*; *Feng et al., 2020*).

Here, we focused on identifying additional roles of NRL in guiding rod gene expression by characterizing NRL-interacting proteins from the mammalian retina using multiple complementary approaches. We discovered an over-representation of RNA-binding proteins (RBPs) among NRL interactors, with almost half of the high-confidence interacting proteins associated with R-loops, which are DNA structures comprised of RNA–DNA hybrids and displaced single-stranded DNA. We then investigated the interaction of NRL with two key RNA helicases, DHX9 and DDX5, which mediate resolution of R-loops and further evaluated R-loop dynamics and epigenetic signatures in the retina. Our study underscores the importance of RBPs and R-loop formation as key NRL-interacting regulators of gene expression in retinal rod photoreceptors.

## Results

### RBPs constitute a large cohort among NRL interactors

To identify new NRL protein partners that mediate its function in rod photoreceptors, we first conducted an unbiased protein interaction screening. This involved a GST-NRL affinity purification using bovine retina nuclear extracts. This strategy was followed by co-immunoprecipitation (IP) from NRL-enriched high molecular mass fractions of bovine retina, NRL co-immunoprecipitation (co-IP) from mouse retina, and yeast-two-hybrid (Y2H) assays with an NRL domain bait against a human retina 'prey' library (schematically shown in *Figure 1A*).

We identified 28 proteins that were significantly enriched (>twofold, p < 0.05) compared to control samples in three independent GST-NRL pull-down experiments (*Figure 1B*, *Supplementary file 1*). Most proteins were not detected in any of the GST control samples. RBPs represented over 40% of NRL interactors in this set (12/28) (*Supplementary file 1*). We confirmed six of the proteins exhibiting the strongest signals by immunoblot analysis of GST-pull-down proteins using specific antibodies (*Figure 1B*). Notably, interaction of NRL with DHX9, HNRNPU, or HNRNPUl1 remained robust even after incubation of the nuclear lysate with Benzonase, a promiscuous nuclease that digests all nucleic acid species, whereas HNRNPA1, HNRNPM, and HNRPA2B1 were not or barely pulled down after treatment, indicating a requirement of nucleic acids for the latter set of interactions (*Figure 1—figure*

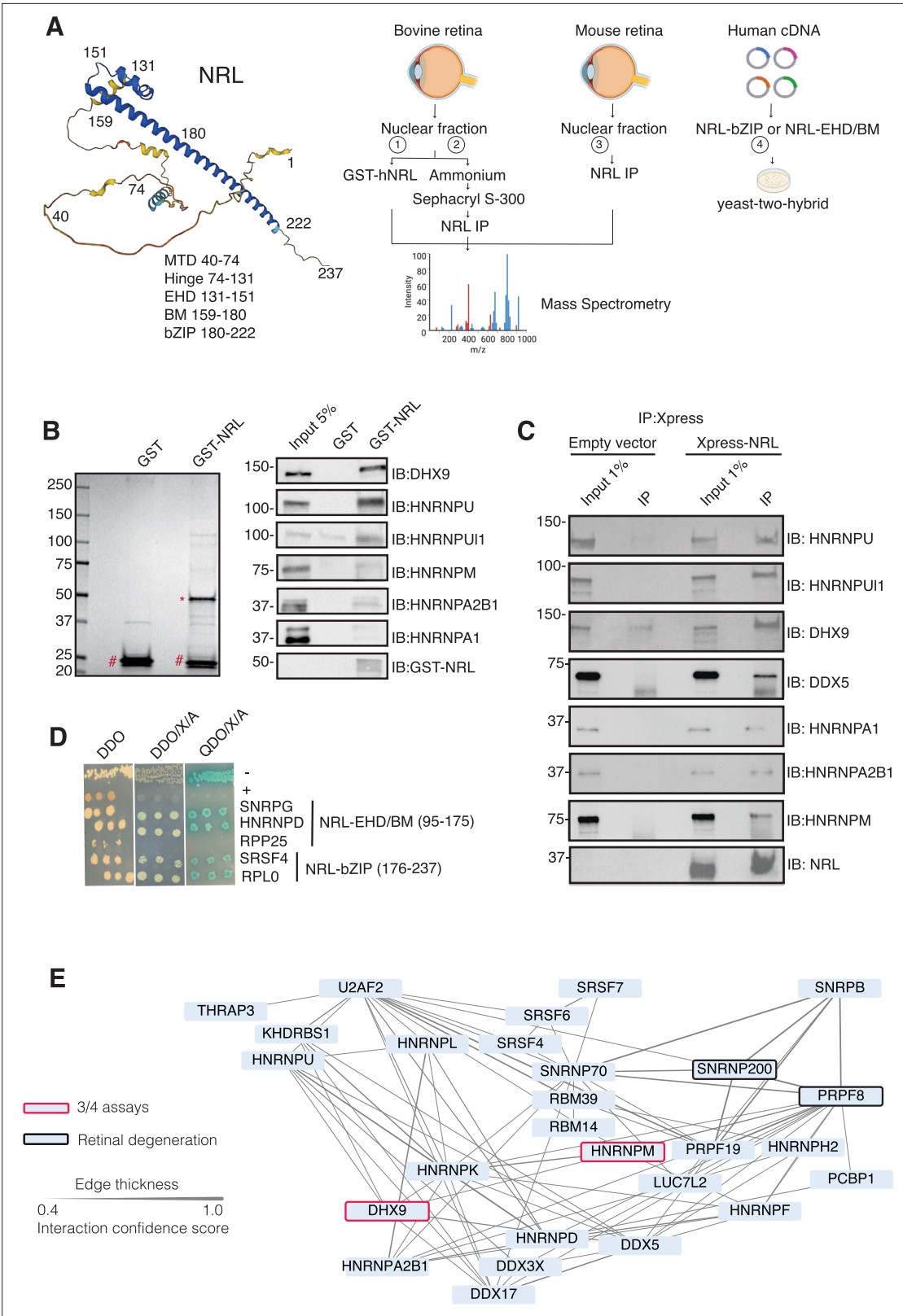

**Figure 1.** Neural retina leucine (NRL) interacts with RNA-binding proteins (RBPs). (**A**) Summary of experimental strategies used to identify NRL interactors. An AlphaFold-predicted model of human NRL is shown, with protein domains highlighted. MTD = minimal transactivation domain, EHD = extended homology domain, BM = basic domain, bZIP = basic leucine zipper. Four different assays were performed to identify NRL interactors. Affinity purifications with Glutathione and NRL antibodies were performed from bovine and mouse retinal lysates and subjected to mass spectrometry. Yeast-

*Figure 1 continued on next page*

*Figure 1 continued*

two-hybrid experiments from human retina cDNA using NRL bZIP and EHD/BM domains were also performed. Eye, plasmid and yeast depictions were obtained from BioRender. (**B**) Coomassie staining showing proteins from bovine retina purified with human NRL fused to GST (GST-NRL; *). Purified GST (#) was used as control. Experiments were performed three times with different retinal lysates. Western blot of RBPs identified by LS/MS harboring >10 times enrichment in at least one GST-NRL replicate compared to controls is shown to the right. (**C**) Western blot showing detection of different RBPs co-immunoprecipitating with NRL in HEK293 cells overexpressing Xpress-tagged NRL. Empty vector containing Xpress tag was used as control. (**D**) Yeast colonies from yeast-two-hybrid screens showing positive interaction between RBPs and NRL extended homology domain (EHD) and basic leucine zipper (bZIP) domain. Colonies were plated against controls on SD/-Leu/-Trp (Double Dropout; DDO), SD/-Trp/-Leu/X-alpha-gal/Aureobasidin-A (DDO/X/A) and SD/-Trp/-Leu/-Ade/-His/X-alpha-gal/Aureobasidin-A (QDO/X/A) plates. P53 and Lamin were used as positive and negative controls, respectively. (**E**) Protein–protein interaction (PPI) network showing RBP experimental interactions from String. Proteins represent a subnetwork of NRL-interacting RBPs found in two out of four assays summarized in A. The edge thickness represents the confidence score with a cutoff of 0.4. Proteins identified in three out of four assays are highlighted with a red border. Proteins with known causative mutations for inherited retinal degeneration are shown with a black border.

The online version of this article includes the following source data and figure supplement(s) for figure 1:

**Figure supplement 1.** Neural retina leucine–RNA-binding protein (NRL–RBP) interactors are enriched in R-loop proteins.

**Figure supplement 1—source data 1.** PDF file containing original western blots for *Figure 1B, C*, *Figure 1—figure supplement 1A* indicating the relevant bands.

**Figure supplement 1—source data 2.** Original files for western blots displayed in *Figure 1B, C*, *Figure 1—figure supplement 1A*.

*supplement 1A*). Removal of nucleic acids significantly increased the background binding for most proteins; however, GST-NRL always maintained stronger signals than controls. This includes HNRNPU and HNRNPUL1 that display white bands indicative of substrate depletion due to high protein levels (*Figure 1—figure supplement 1A*).

Independently and concurrently, immunoprecipitation using anti-NRL antibody of NRL-enriched high molecular mass fractions from the bovine retina, followed by mass spectrometry analysis, identified several RBP interactors in addition to known interacting proteins such as CRX, NR2E3, and NonO/p54$^{nrb}$ (*Supplementary file 2*). Prioritization of RBPs was done by selecting those with a combined peptide spectrum match (PSM) higher than 10 and at least twofold enrichment. Notably, the most enriched proteins included DHX9, HRRNPU, and HNRNPM. Additionally, NRL co-IP experiments using mouse retina nuclear extracts followed by mass spectrometry further validated these findings (*Supplementary file 2*). Out of the 786 and 1491 proteins identified in bovine and mouse pull downs, respectively (*Supplementary file 2*), 'mRNA-binding' was the top 4% highest enriched gene ontology term (adjusted p-value = 1.27E−53 and 2.1E−21, respectively). We also confirmed the interaction between NRL and several of the identified proteins by transfection of tagged constructs in HEK293 cells overexpressing human NRL (*Figure 1C*). A limited Y2H screening using the NRL bZIP or extended homology domain (EHD) domain baits also identified several RBPs, including SNRPG, HNRNPD, SRSF4, and RPLP0, which were confirmed by positive bait and prey interactions (blue color) (*Figure 1D*).

Candidate NRL-interacting RBPs, obtained from all four assays (*n* = 197), were annotated with curated protein–protein interactions (PPIs) obtained from String (https://string-db.org/) *Szklarczyk et al., 2023* to generate a PPI network (*Figure 1—figure supplement 1B*). This network displayed highly interconnected nodes that may indicate interactions with NRL in different cellular contexts. Notably, 30 of these proteins were identified in at least two assays, and two of these, DHX9 and HNRNPM, in three assays (*Figure 1—figure supplement 1B*). Of these, 27 RBPs show a high degree of interaction and form a highly interconnected subnetwork (*Figure 1E*). Notably, two of the RBPs (PRPF8 and SNRNP200) *Escher et al., 2018* have been implicated in photoreceptor degeneration and harbor substantial interactions in this subnetwork (*Figure 1E*, black rectangle). We also noted that some RBPs in the network (DHX9, DDX5, DDX17, and DDX3X) are RNA helicases with prominent roles in the regulation of R-loops, which are non-B DNA structures comprising of RNA-DNA hybrids and displaced single-stranded DNA with multiple regulatory roles in transcription (*Kim and Wang, 2021*). We then identified 67 high-confidence R-loop binding proteins from four independent studies (*Cristini et al., 2018*; *Li et al., 2020*; *Mosler et al., 2021*; *Wang et al., 2018*) by determining shared proteins in their respective R-loop proteomes. Strikingly, 50% of NRL-interacting RBPs were also identified in this high-confidence R-loop protein group indicating a high enrichment of R-loop-associated

RBPs among NRL interactors (Fisher's exact test p = 2E−13) (*Figure 1—figure supplement 1C*, *Supplementary file 3*).

## NRL interacts with the R-loop helicases DHX9 and DDX5 in rod photoreceptor nuclei

We further focused on the interaction of NRL with R-loop resolvases DHX9 (identified in all affinity purifications) and DDX5 (identified in endogenous NRL-pull downs from bovine and mouse retinas). DHX9 and DDX5 are shown to be present in the same protein complexes (*Padmanabhan et al., 2012*). Notably, immunofluorescence of mouse retina reveals the localization of DHX9 and DDX5 in the rod nuclear periphery, which corresponds to the euchromatin compartment of murine rods (*Figure 2—figure supplement 1A*). We therefore studied DHX9 and DDX5 interaction with NRL in situ using a proximity ligation assay (PLA). In mouse retina, DHX9 is observed in all nuclei (*Figure 2A*). PLAs using anti-NRL and DHX9 antibodies show robust positive interaction signal in the outer nuclear layer (*Figure 2B*) where NRL is normally expressed (*Figure 2A*). In contrast, no signal above background is detectable when PLA is performed in *Nrl* KO retina (*Figure 2B*), suggesting the specificity of PLA signals, even though DHX9 is still expressed throughout the KO retina nuclei (*Figure 2A*). DDX5 also shows widespread expression throughout the retina (*Figure 2C*) but displays a lower yet specific interaction signal with NRL in the outer nuclear layer by PLA (*Figure 2D*). This signal is absent in *Nrl* KO retina (*Figure 2D*). Similar to mouse retina, DHX9 and DDX5 are detected in all nuclear layers of the human retina whereas NRL is present only in the outer nuclear layer of photoreceptors (*Figure 2E, G*). Consistent with the results in mouse, NRL–DHX9 PLA signal is clearly detected in the outer nuclear layer of the human retina (*Figure 2F*), indicating a robust and reproducible interaction between NRL and DHX9. However, NRL-DDX5 PLA signals do not seem to be above negative control (*Figure 2H, I*). To further ensure that the signal is specific for DHX9, negative controls using human retinas were performed with DHX9 antibodies in combination with Goat IgG. Some background is observed outside the nucleus and in the extracellular space, but no signal enrichment is detected in the photoreceptor nuclear layer (*Figure 2—figure supplement 1B*). Interestingly, in HEK293 cells overexpressing human NRL, PLA signals are robust for both DHX9 and DDX5 throughout the nuclear compartment (*Figure 2J*). DHX9 and HNRNPU serve as a positive control for PLA and are enriched mostly in the nuclear periphery (*Figure 2J*). Taken together, these findings suggest a strong interaction between NRL and DHX9 throughout the nuclear compartment in the retina and that a transient and/or more regulated interaction of NRL with DDX5 may require additional protein partners.

## NRL regulates *DHX9* gene expression

As transcription factors often operate on positive feedback loops, we studied whether the gene expression of NRL interactors was influenced by NRL itself. We examined the regulation of NRL-interacting RBPs by inspecting the published NRL ChIP-Seq and super enhancer profiles from the human retina (*Marchal et al., 2022*). Interestingly, over 50% of RBP genes possess NRL ChIP-Seq peaks and almost 25% harbor super enhancers (SEs) (*Figure 3—figure supplement 1A*). *Figure 3—figure supplement 1A* shows a few examples of NRL binding to super enhancers regions at *DDX5*, *HNRNPU*, *DDX3X*, and *THRAP3* genes. We then focused on the *DHX9* locus and could identify NRL-binding peaks in both human and mouse promoter regions (*Figure 3B*, *Figure 3—figure supplement 1B*). In concordance, we observe lower levels of *Dhx9* transcripts in *Nrl* KO photoreceptors (*Kim et al., 2016*; *Figure 3C*). A search of NRL ChIP-Seq peaks (*Hao et al., 2012*) at the human *DHX9* promoter identified a canonical NRL-binding site that is partially conserved in mouse, in concordance with NRL Cut&Run data from mouse retina (*Liang et al., 2022*; *Figure 3D*). Using a probe containing this sequence, we demonstrate binding of bovine retina nuclear proteins by electrophoretic mobility shift assays (EMSAs) (*Figure 3D*). Although we did not directly measure NRL binding to this sequence, probes with mutations in and around the putative NRL-binding site were not able to compete with the wild-type probe for binding (*Figure 3E*). Taken together, our data suggest that NRL can bind to and influence the expression of *Dhx9* and probably other genes encoding RBP interactors.

## Interaction between NRL and DHX9 is regulated by R-loops

We further characterized whether the interactions between NRL and DHX9 or DDX5 are regulated by different nucleic acids in HEK293 cells and/or bovine retina. We therefore performed co-IP

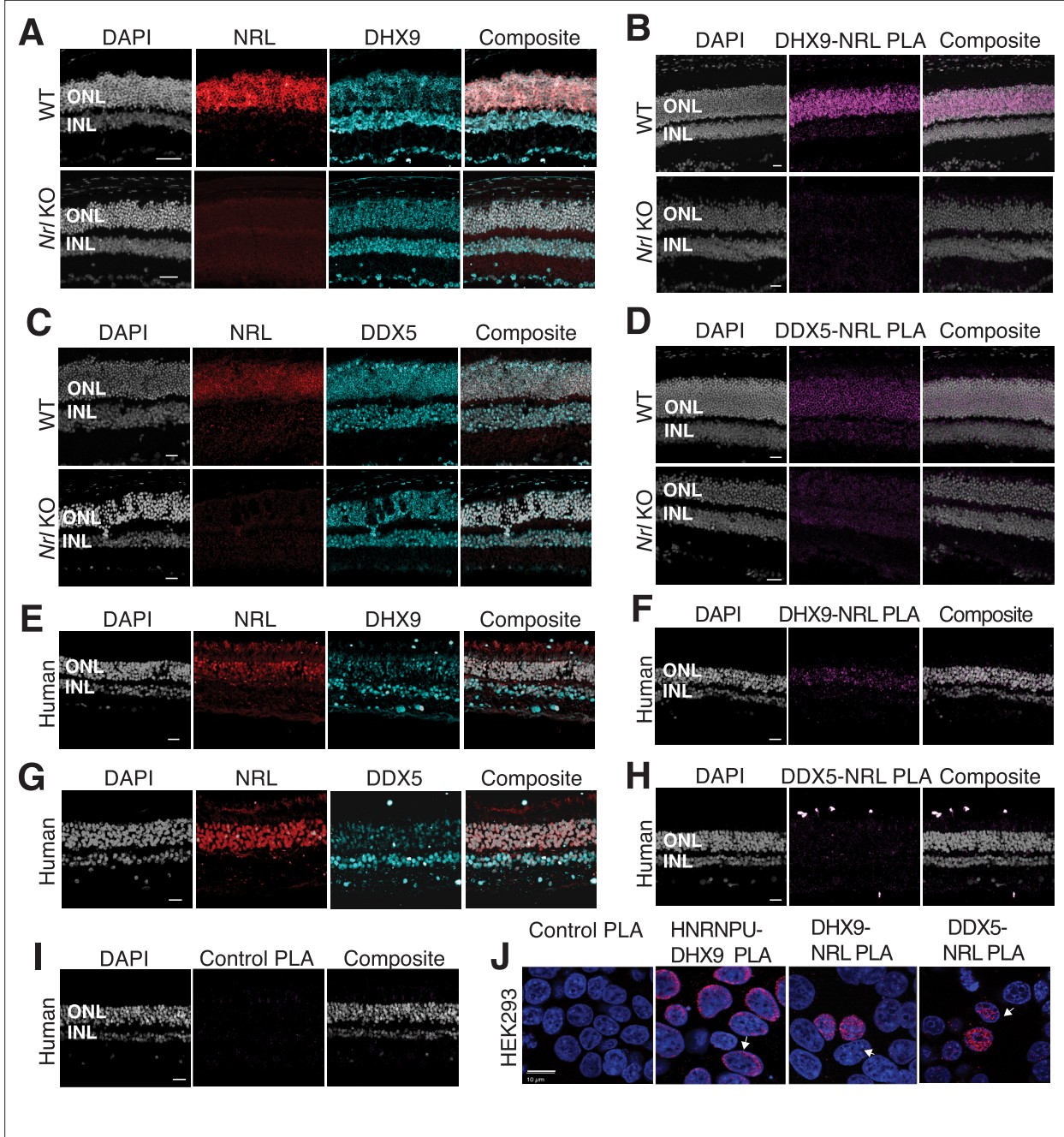

**Figure 2.** DHX9 and DDX5 are expressed in retinal photoreceptors and interact with neural retina leucine (NRL) within the nuclear compartment. (**A**) DHX9 (blue) and NRL (red) expression in adult mouse wild-type (WT) and *Nrl* Knockout (KO) retina. (**B**) Proximity ligation assay (PLA) signal (magenta) using anti-DHX9 and NRL antibodies in adult mouse WT and *Nrl* KO retina. (**C**) DDX5 (blue) and NRL (red) expression in adult mouse WT and *Nrl* KO retina. (**D**) PLA signal (magenta) using anti-DDX5 and NRL antibodies in adult mouse WT and *Nrl* KO retina. (**E**) DHX9 (blue) and NRL (red) expression in adult human retina. (**F**) PLA signal (magenta) using anti-DHX9 and NRL antibodies in the adult human retina. (**G**) DDX5 (blue) and NRL (red) expression in adult human retina. (**H**) PLA signal (magenta) using anti-DDX5 and NRL antibodies in the adult human retina. (**I**) PLA signal (magenta) using no primary antibody in the adult human retina. (**J**) PLA signal in HEK293 cells overexpressing human Xpress–NRL. DHX9 interaction with its known protein partner HNRNPU is shown in the nuclear periphery (arrow). Xpress–NRL interaction with DDX5 and DHX9 in euchromatin is shown in red (arrows). Nuclei were counterstained with DAPI (gray in human and mouse retina; blue in HEK293 cells). Scale bar = 20 µM. ONL = outer nuclear layer; INL = inner nuclear layer.

The online version of this article includes the following figure supplement(s) for figure 2:

**Figure supplement 1.** Subcellular localization of DHX9 and DDX5 in adult mouse retina (P28).

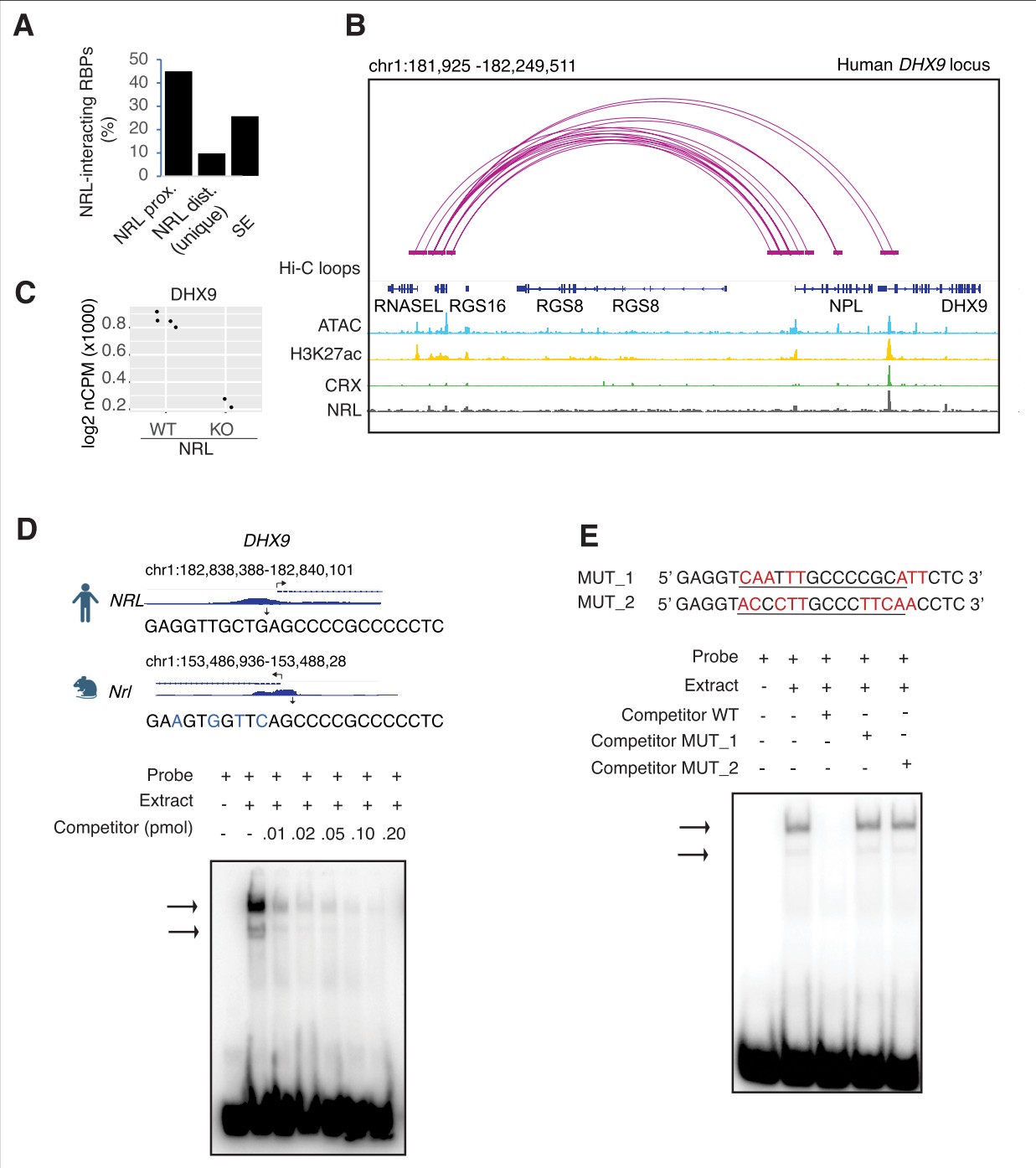

**Figure 3.** Neural retina leucine (NRL) genetically interacts with RNA-binding proteins (RBPs). (**A**) Bar graph showing the fraction of NRL-interacting RBPs that harbor NRL proximal (±1 kb gene body) or distal (>1 kb gene body) ChIP-Seq peaks and/or super enhancers (SEs) in the human retina (data from *Marchal et al., 2022*). (**B**) Genomic view of the human *DHX9* locus showing Hi-C loops, ATAC-Seq, H3K27ac, CRX-ChIP-Seq, and NRL-ChIP-Seq tracks (obtained from *Marchal et al., 2022*). (**C**) Expression levels of *Dhx9* in wild-type and *Nrl* knockout flow-sorted photoreceptors (obtained from *Kim et al., 2016*). (**D**) Electrophoretic mobility shift assay (EMSA) autoradiography using a probe containing an NRL motif identified at NRL-ChIP-Seq peak on the human DHX9 promoter. Chip-Seq or CUT&RUN profiles for human and mouse NRL at the *DHX9* promoter (obtained from *Marchal et al., 2022* and *Liang et al., 2022*, respectively) are shown. The location of the NRL motif is shown with arrows. The sequence of the $^{32}$P-labeled probe containing human NRL motif (underlined) and its homologous sequence in mouse is shown on the top panel (blue letters indicate nucleotide differences). The blot depicts the formation of specific bands (arrows) after incubation with bovine nuclear retina extracts. Competition assays were performed using unlabeled probes at increasing concentrations (pmol) as shown. (**E**) EMSA autoradiography showing competition assays with 0.2 pmol WT and mutant DHX9 probes, MUT_1 and MUT_2 (sequences are shown in top panel; nucleotide changes are shown in red).

*Figure 3 continued on next page*

*Figure 3 continued*

The online version of this article includes the following figure supplement(s) for figure 3:

**Figure supplement 1.** Neural retina leucine (NRL) occupancy on super enhancers at genes encoding NRL-interacting RNA-binding proteins (RBPs).

experiments with antibodies against NRL, DHX9, and DDX5 with or without RNase A, RNase H, and DNase. IP using a DDX5 antibody failed to consistently pull down NRL from HEK293 and bovine retina extracts, suggesting that only a small pool of NRL is involved in this interaction, or their interaction is weak and transient. In HEK293 cells overexpressing NRL, the interaction of DHX9 and NRL is enhanced upon treatment of lysates with RNase A (*Figure 4A, C*), indicating a negative regulation by RNA. We observed a similar phenomenon in reciprocal pull downs with DHX9 (*Figure 4B, D*). On the other hand, treatment with RNase H, which digests RNA only in the DNA:RNA hybrid context, reduced NRL's interaction with both DHX9 and DDX5 suggesting a positive impact of DNA:RNA hybrids (*Figure 4A, C*). The reciprocal pull downs with DHX9 did not reveal this change (*Figure 4B, D*) probably since NRL accounts for only a small part of DHX9 interactome. Pull down of DHX9 and DDX5 from bovine retina extract with endogenous NRL was not altered by RNase A treatment but was decreased upon removal of DNA:RNA hybrids (*Figure 4E, G*), suggesting a role of R-loops in regulating NRL interactions. As in HEK293 cells, reducing DNA:RNA hybrids did not affect the amount of NRL that was pulled down from bovine retina with DHX9 antibody (*Figure 4D, F*). Notably, the binding between DHX9 and DDX5 was decreased when RNase A was present (*Figure 4F*, *Figure 4—figure supplement 1A*), indicating a positive regulatory role of RNA in the retina. Yet, the binding between DHX9 and DDX5 was reduced by RNase H treatment in HEK293 cells, but not in the bovine retina as observed in case of NRL interactions (*Figure 4B, F*, *Figure 4—figure supplement 1B*). DNA did not influence NRL interactions as evidenced by treatments of lysates with DNase I, which has only 1% activity on DNA:RNA hybrids (*Figure 4A–D*).

To assess whether NRL and DHX9 bind to retinal R-loops, we isolated R-loops from enzymatically digested retina gDNA and performed DNA–RNA immunoprecipitation (DRIP) using the S9.6 antibody. After incubations with mouse retinal nuclear lysates, NRL is enriched in R-loops along with DHX9 (*Figure 5A*). A pool of NRL also binds to the RNase H-treated sample, possibly reflecting its ability to interact with genomic DNA sequences that may still be present in the pull down. However, DDX5 does not appear to bind to retinal R-loops, suggesting that DDX5 association with R-loops is weaker, transient, or more regulated in the retina, and in agreement with co-IP findings from RNase H-treated bovine retina lysates. We then overexpressed GFP-tagged RNase H1 in HEK293 cells to reduce the R-loop levels, as shown previously (*Crossley et al., 2023*), and co-transfected the cells with human NRL. We detect a reduction in the number of cells displaying NRL–DHX9 nuclear PLA signals in cells that also harbor wild-type nuclear RNase H1 as compared to the cells having catalytically inactive version of RNase H1 (GFP-dHR) (*Figure 5B and D*). In addition, we observe a small percentage of cells with PLA signals enriched in subnuclear compartments, especially in cells with wild-type RNase H1 (*Figure 5B, E*, arrows). Notably, the localization of NRL or DHX9 is not influenced by the expression of RNase H1 (*Figure 5C*, *Figure 5—figure supplement 1C*). Thus, NRL and DHX9 interaction in the nucleoplasm is favored by R-loops, and R-loops appear to influence the stabilization and localization of NRL–DHX9 complexes.

## R-loops are enriched in neuronal genes and display distinct epigenetic signatures

NRL's interplay with R-loops and R-loop proteins suggests a yet unexplored role of R-loops in regulating retinal transcriptional programs. Quantification of R-loop levels using the S9.6 antibody reveals an increase in R-loops with postnatal retinal maturation (*Figure 6A*). This is in contrast with the relatively stable levels of total expressed genes (~15,000 > 1 count per million (CPM)) and transcripts (~35,000 > 1 RPKM (Reads Per Kilobase per Million mapped reads)) during retinal development (*Brooks et al., 2019*). Total R-loop levels increase in the *Nrl* KO retina, indicating a role of R-loops in controlling cell type-specific gene expression and/or the role of NRL in regulating R-loop levels (*Figure 6—figure supplement 1*).

To identify genomic elements associated with R-loops, we performed single-strand DRIP (ssDRIP) followed by sequencing with four independent adult mouse retina samples and used corresponding

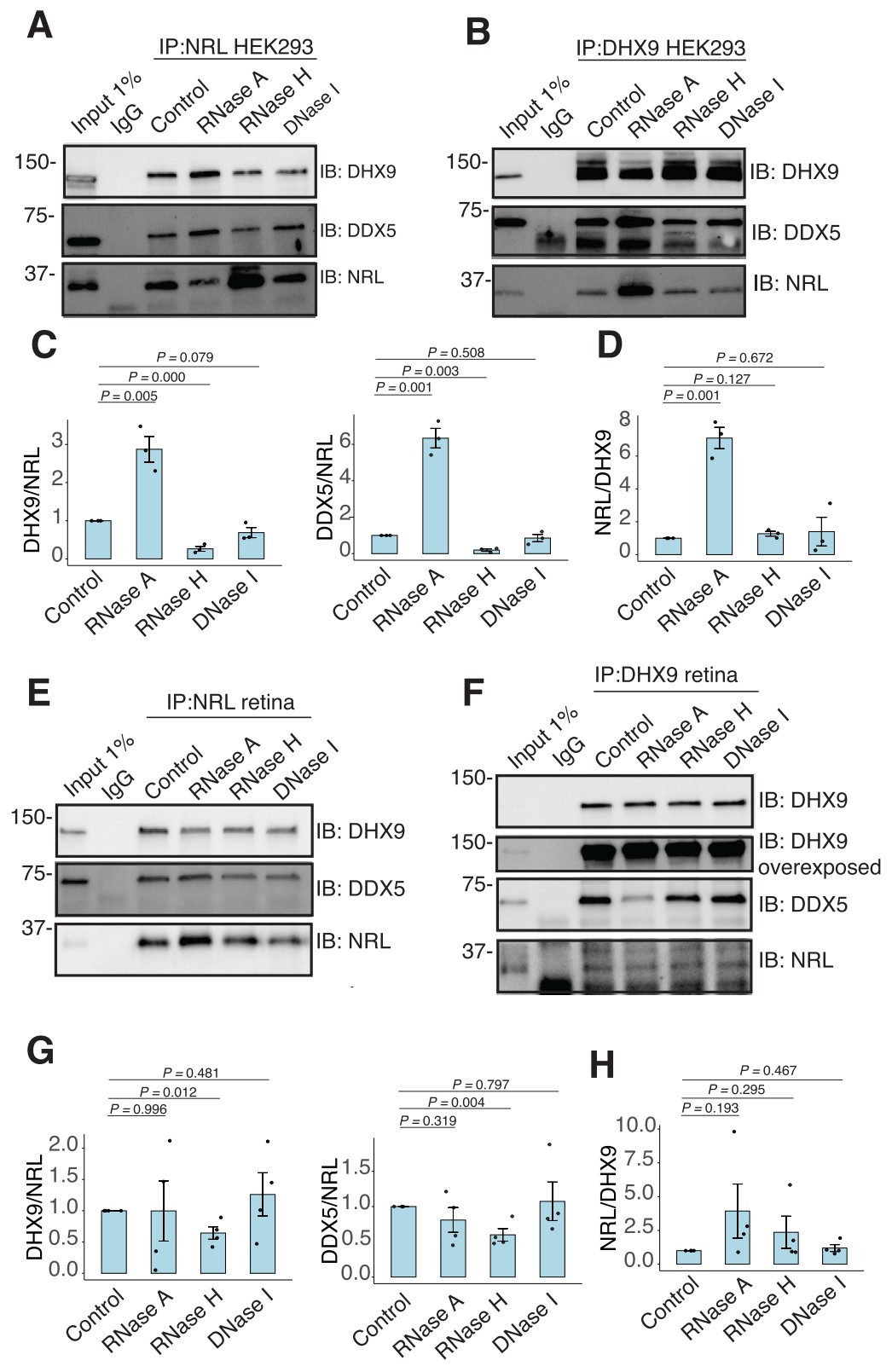

**Figure 4.** RNA:DNA hybrids regulate the interaction between neural retina leucine (NRL) and DDX5/DHX9. (**A, B**) Co-immunoprecipitation (co-IP) of DDX5 and DHX9 from HEK293 cells overexpressing NRL. Lysates were treated for 30 min with different nucleases (as shown) before incubations with respective antibodies. Immunoprecipitation (IP) of NRL (**A**) or DHX9 (**B**) and immunoblot (IB) staining for NRL, DHX9, and DDX5 is shown. (**C, D**) Quantification

*Figure 4 continued on next page*

*Figure 4 continued*

of signal intensities normalized to affinity-purified NRL and DHX9 (shown in A and B, respectively) (*n* = 3). Data are presented as the mean ± SEM. An unpaired two-tailed *t*-test was performed to compare the means of samples against controls. (**E, F**) Co-IP of DDX5 and DHX9 from nuclear fractions of bovine retinas. Lysates were treated for 30 min with different nucleases (as shown) before incubations with respective antibodies. IP of NRL (**E**) or DHX9 (**F**) and immunoblot (IB) staining for NRL, DHX9, and DDX5 is shown. (**G, H**) Quantification of signal intensities normalized to affinity-purified NRL and DHX9 (shown in E and F, respectively) (*n* = 4). Data are presented as the mean ± SEM. Unpaired two-tailed *t*-test was performed to compare means of samples against controls.

The online version of this article includes the following source data and figure supplement(s) for figure 4:

**Source data 1.** PDF file containing original western blots for *Figure 4A, B, E, F*, indicating the relevant bands and treatments.

**Source data 2.** Original files for western blots displayed in *Figure 4A, B, E, F*.

**Figure supplement 1.** DHX9 interacts with DDX5 in HEK293 and bovine retina.

RNAse H-treated controls. Remarkably, all four samples and all four RNAse H-treated samples clustered separately by their PC1 value (*Figure 6—figure supplement 2A*, Principal component analysis (PCA)), which contributes to 58.12% of the variance whereas PC2 contributes to 16.26%, demonstrating the specificity of the assay. We identified 4677 R-loops in our ssDRIP-Seq data that were absent in RNase H-treated samples (*Figure 6—figure supplement 2B, C*). We assessed the strand specificity of these R-loops by quantifying the enrichment of the ssDRIP-Seq signal over the opposite strand (*Figure 6B*, *Figure 6—figure supplement 2B*). Our analysis uncovered 3352 loops without any strand specificity (hereafter referred as unstranded) and 1328 strand-specific R-loops (hereafter referred as stranded). Notably, the DRIP-Seq coverage was similar for both unstranded and stranded R-loops (*Figure 6—figure supplement 2C*). The identified R-loops intersected with different intergenic and genic regions and are overrepresented in intergenic and promoter regions (*Figure 6C*, *Figure 6—figure supplement 2D*). Intriguingly, stranded R-loops are also enriched at pseudogenes, whereas unstranded R-loops are only enriched at intergenic regions (*Figure 6C*). In agreement, stranded R-loops are preferentially observed around the transcription start site (TSS), but unstranded R-loops are further downstream of the transcription termination site (*Figure 6—figure supplement 2E*).

We then studied whether certain genes/pathways display more R-loops in the retina. We found that stranded R-loops were particularly enriched at neuronal genes associated with synapse function (*Figure 6D*) and unstranded R-loops were also enriched at genes associated with G-protein-coupled receptor signaling (*Figure 6—figure supplement 2F*). In addition, R-loops were identified in 20 genes involved in retinal disease including *Ush2a*, *Pcdh15*, and *Abcc6* (*Supplementary file 4*).

Subsequently, we examined the association of R-loops with different chromatin marks using published retina datasets (*Norrie et al., 2019*; *Figure 6E*). We observed that stranded and unstranded R-loops displayed differences in their epigenetic signatures. Stranded R-loops harbor histone marks associated with active transcription including H3K36me3, H3K4me2, and H3k4me3. However, unstranded R-loops were only enriched with H3K9me3, associated with heterochromatin (*Figure 6E*). Next, we investigated whether NRL binds to R-loops and found that NRL is enriched at both stranded an unstranded R-loops, suggesting a strong association of NRL to R-loops regardless of the epigenetic context (*Figure 7A*). Notably, other chromatin-binding factors such as BRD4, CTCF, and Pol II were primarily associated with stranded R-loops. In addition, NRL binding was associated with a higher proportion of R-loops at low and high expression genes (*Figure 7B*, *Figure 7—figure supplement 1A*). We then looked specifically at the NRL-regulated genes (*Liang et al., 2022*; *Supplementary file 5*). Overall, NRL target genes have an enrichment of stranded R-loops at the promoter/TSS and unstranded R-loops on the gene body compared to all genes annotated in Ensembl (*Figure 7—figure supplement 1B*). Select examples of genes displaying NRL occupancy and harboring stranded or unstranded R-loops are shown in *Figure 7C*. R-loops can also overlap with proximal promoter or intergenic regions that do not show NRL binding (*Figure 7—figure supplement 1C*). Conversely, R-loops are undetectable in few highly expressed NRL-regulated genes such as *Rhodopsin* (*Figure 7—figure supplement 2*). Finally, we studied whether R-loop formation is linked to the gene length. Curiously, the genes without any R-loop are shorter than those overlapping with R-loops, independently of their expression level (*Figure 7D*). In genes showing low to mid-level expression, a concomitant formation

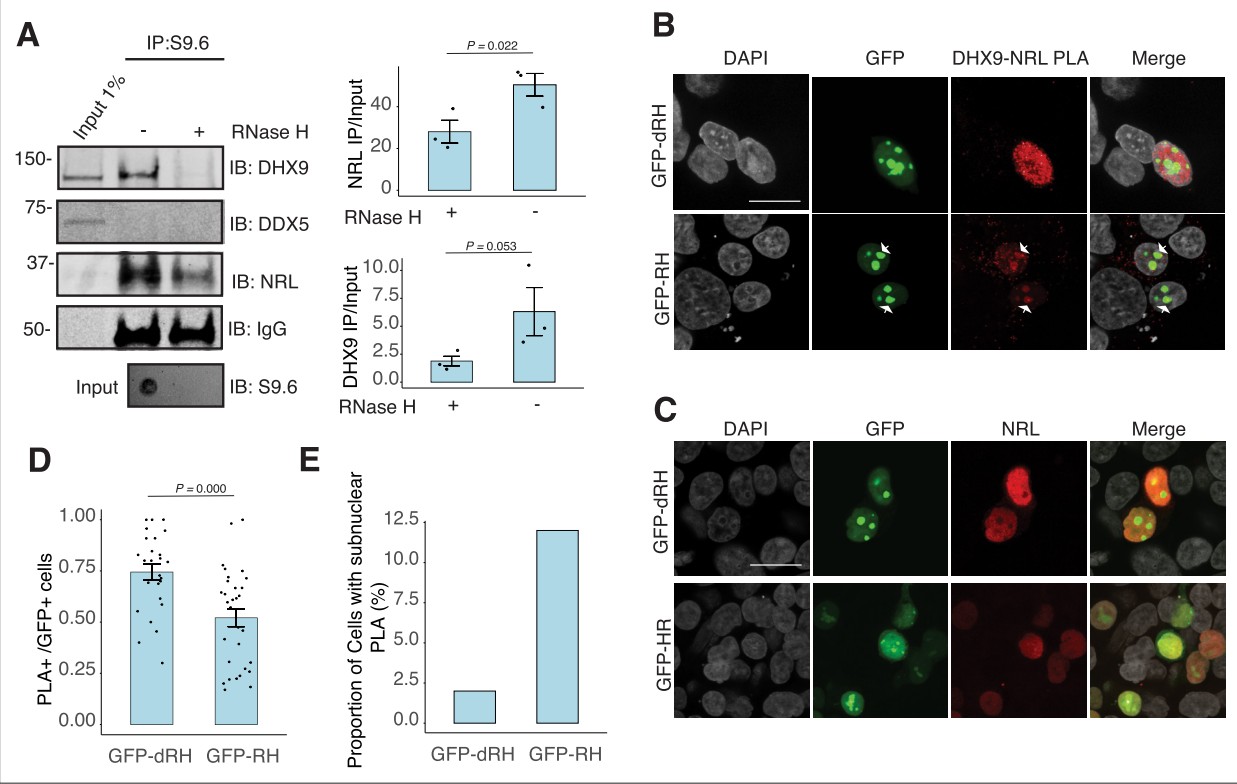

**Figure 5.** Nuclear R-loops regulate the interaction between neural retina leucine (NRL) and DHX9. (**A**) Western blot of DNA:RNA hybrid immunoprecipitation (DRIP) from adult mouse retina showing immunoblot (IB) staining for DHX9, NRL, and DDX5. Retinal genomic DNA (gDNA) was digested with MseI, DdeI, AluI, MboI, incubated with RNase III with/without RNase H and immunoprecipitated with S9.6 antibody/protein G beads. Retinal nuclear lysates were incubated with antibody/bead complexes. Quantification of signal intensities of immunoprecipitated DHX9 and NRL compared to input (*n* = 3). Data are presented as the mean ± SEM. Unpaired one-tailed *t*-test was performed to compare means of samples against controls. (**B**) Confocal image of HEK293 cells transfected with NRL and wild-type (WT) human RNase H1 or D201N catalytic dead mutant EGFP fusions (GFP-HR and GFP-dHR, respectively). Proximity ligation assay (PLA) signals using antibodies for NRL and DHX9 are shown in red. Some cells displayed nucleolar-like accumulation of PLA signal (arrows). (**C**) Confocal image of HEK293 cells transfected with NRL and GFP-dRH or GFP-RH and stained with antibodies against NRL (red). Nuclei are stained with DAPI (gray). Scale bar is 20 μM. (**D**) Quantification of cells with positive PLA signals from B. Each dot represents a ratio of number of GFP+ cells with nuclear PLA signals per image. Data are presented as the mean ± SEM. Unpaired two-tailed *t*-test was performed to compare means of samples against controls. (**E**). Bar graph showing percentage of EGFP+ cells harboring NRL–DHX9 PLA signals in subnuclear compartments from B. Cells were counted in four independent assays (*n* = 83 and 85 cells for GFP-dHR and GFP-RH, respectively).

The online version of this article includes the following source data and figure supplement(s) for figure 5:

**Source data 1.** PDF file containing original western blots for *Figure 5A*, indicating the relevant bands.

**Source data 2.** Original files for western blots displayed in *Figure 5A*.

**Figure supplement 1.** Expression of DHX9 in RNase H1-overexpressing HEK293 cells.

of stranded and unstranded R-loops is associated with longer genes compared to the genes with stranded only or unstranded only R-loops (*Figure 7D*). Taken together, these data demonstrate distinct associations between R-loop types, genomic features, and chromatin-binding factors including NRL, which could play a role in the regulation of cell type-specific gene expression.

## Discussion

RBPs are part of numerous nucleoprotein complexes that participate in diverse cellular processes including transcription of genes in appropriate spatiotemporal context (*Van Nostrand et al., 2020*; *Gebauer et al., 2021*). Several RBPs interact with chromatin and function as integrators of transcription and co-transcriptional RNA processing (*Xiao et al., 2019*). We currently have poor understanding of how ubiquitously expressed RBPs are recruited to specific genomic loci at defined locations and in distinct cell types. Here, we demonstrate the interaction of retinal rod photoreceptor-specific bZIP

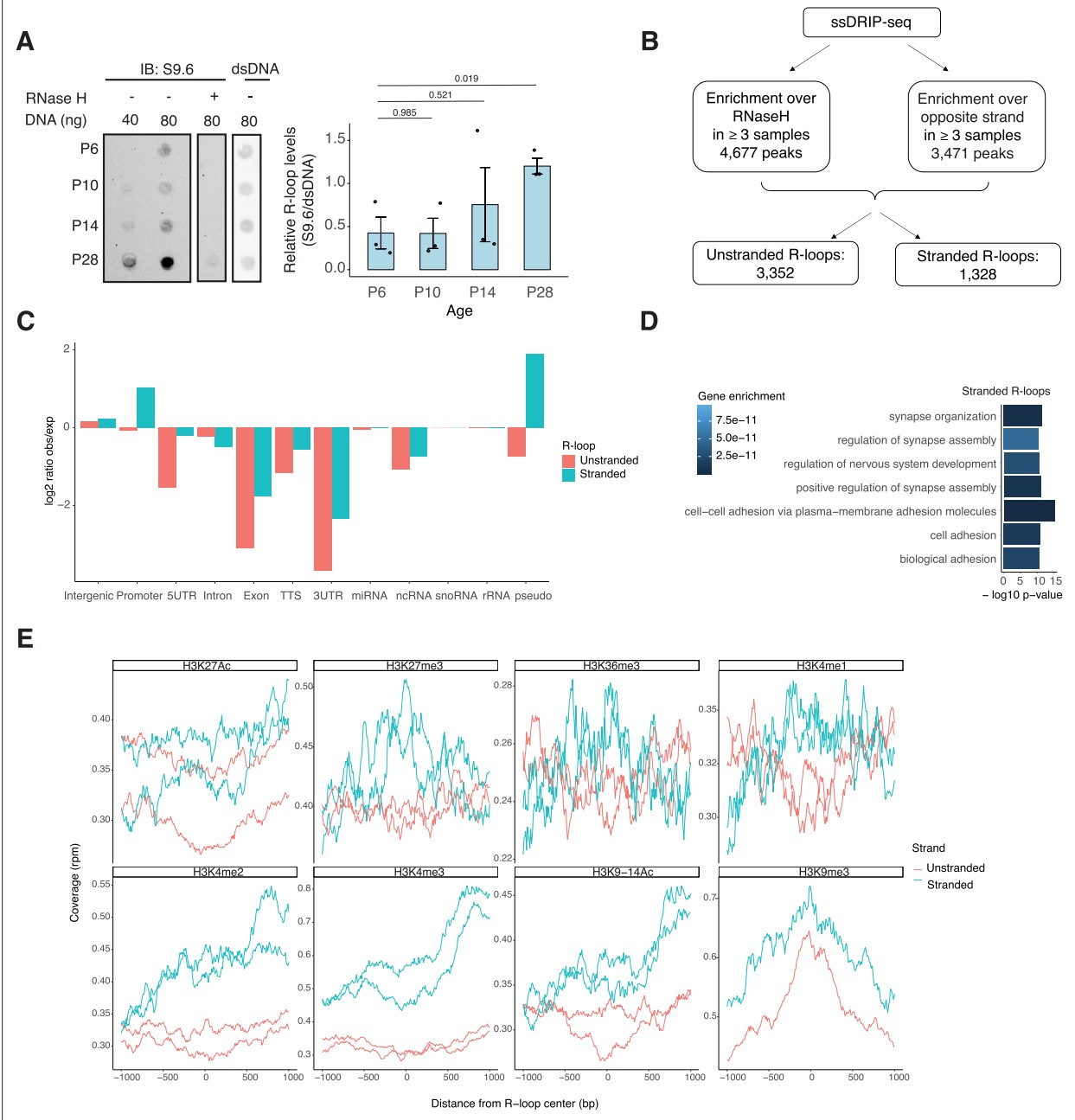

**Figure 6.** R-loops are dynamic in the mouse retina and associate with distinct epigenetic signatures. (**A**) Dot blot of DNA:RNA hybrids from retinal gDNA. Retinas were dissected from mice at different developmental stages as shown. Genomic DNA (gDNA) was treated with RNaseIII with and without RNase H overnight. R-loops were detected using S9.6 antibody (*n* = 3). Data are presented as the mean ± SEM. Unpaired two-tailed *t*-test was performed to compare means of samples against controls. (**B**) R-loop peaks from ssDRIP-Seq were identified using RNase H-treated samples as controls. R-loops found in at least three samples with a *q*-value <0.001 in the narrow call or in the broad call were merged. Stranded R-loops were filtered using the opposite strand as reference. (**C**) Observed versus expected ratio of unstranded and stranded R-loops at different genomic regions. (**D**) Biological process enrichment of genes associated with stranded R-loops. (**E**) Metaplot of H3K27ac, H3K27me3, H3K36me3, H3K4me1, H3K4me2, H3K4me3, H3K9-14ac, and H3K9me3 signals centered on stranded and unstranded R-loop peaks.

The online version of this article includes the following source data and figure supplement(s) for figure 6:

**Source data 1.** PDF file containing original dot blots for *Figure 6A*, indicating the relevant treatments.

**Source data 2.** Original files for dot blots displayed in *Figure 6A*.

**Figure supplement 1.** R-loops are increased in neural retina leucine (NRL) KO retina.

**Figure supplement 1—source data 1.** PDF file containing original dot blots for *Figure 6—figure supplement 1*, indicating the relevant treatments.

*Figure 6 continued*

**Figure supplement 1—source data 2.** Original files for dot blots in *Figure 6—figure supplement 1*.

**Figure supplement 2.** Signatures of R-loop formation in the retina.

transcription factor NRL with multitude of RBPs that are associated with RNA splicing as well as resolution of R-loops, which form during the transcription process. Focusing specifically on two helicases – DHX9 and DDX5, we show that their interaction with NRL is dependent on specific nucleic acids, and in the case of DHX9 on R-loops. Notably, R-loops are not uniformly present in genes that are highly expressed in the retina but are detected in heterochromatin regions and in lowly expressed genes. Additionally, R-loops are enriched in genes encoding neuronal synaptic proteins. Our results suggest specific resolution and regulation of R-loops and provide insights into potential mechanisms by which RBPs and R-loops control transcriptional states in the retina.

Most R-loops are formed during transcription and play key roles in transcriptional initiation, elongation, and termination. R-loops are reportedly sufficient to initiate transcription of antisense lncRNA from enhancers and other genomic regions (*Tan-Wong et al., 2019*). R-loops also contribute to Pol II pausing at promoters and in gene bodies, thereby influencing transcriptional outputs (*Tous and Aguilera, 2007*). Accumulation of R-loops at polyadenylation-dependent termination regions (*Sanz et al., 2016*) and R-loop-prone sequences downstream from poly A cleavage sites can cause transcriptional termination (*Skourti-Stathaki et al., 2011*). Additionally, R-loops may indirectly control gene expression by influencing chromatin structure and the epigenome (*Chen et al., 2015*). In this study, we detect enrichment of R-loops at promoter and intergenic regions indicating their potential role in the modulation of distal and internal regulatory elements. We also demonstrate the presence of R-loops in genes irrespective of their expression levels but also linked to their length, suggesting that R-loops are contributors to the complex regulation of long genes.

We are intrigued by the enrichment of R-loops in neuronal genes associated with synaptic structure and function, including those involved in cell–cell communication. As R-loops are generated during transcription, their relatively stable detection at genes encoding several neuronal and retina-disease associated proteins indicates distinct transcriptional control mechanisms for different gene families. One plausible explanation is slower turn-over of RNA and requirement for smaller protein amounts. The greater likelihood of neuronal genes to harbor R-loops may also be because of their larger gene size (*Gabel et al., 2015*). R-loops may also contribute to regulation of long neural genes as is the case for DNA damage (*Wei et al., 2016*). Further investigations into R-loop formation and resolution at distinct genomic regions can offer mechanistic insights into gene regulation as well as non-coding variations relevant to disease.

Our results indicate that unstranded R-loops (containing RNA:DNA hybrids on both strands) are enriched in intergenic regions and occur downstream of genes, overlapping the repressive mark H3K9me3. This observation suggests that bidirectional transcription is associated with the formation of repressive genomic regions. Indeed, R-loops are also reported to be involved in maintenance of heterochromatin and telomers (*Pfeiffer et al., 2013*; *Xu et al., 2017*). The potential role of R-loops in maintaining a repressed chromatin state in the retina warrants further investigation. We note that R-loops near transcriptional termination sites have also been associated with the formation of repressive epigenetic marks and termination of transcription. For example, R-loops can induce antisense transcription at gene termination sequences, leading to the generation of double-stranded RNA and recruitment of G9a histone methyltransferase (*Skourti-Stathaki et al., 2014*). Bidirectional transcription termination by Pol II also offers a potential mechanism of termination at convergent genes (*Wang et al., 2023*). R-loops localizing near H3K9me3 sites are shown to be associated with regulation of long genes (*Manzo et al., 2018*). Intriguingly, NRL genomic occupancy overlaps with unstranded R-loops at the gene body, suggesting a functional link that may be unrelated to transcriptional initiation. Additional experiments are needed to understand the relationship between R-loops and repressive marks in postmitotic retinal neurons.

NRL regulates the transcriptional state of many rod-specific genes (*Kim et al., 2016*). The absolute requirement of NRL for rod photoreceptor cell fate determination and its continued high expression for functional maintenance of rods strongly indicate a multitude of roles of NRL beyond transcription initiation. Interactions of NRL with RBPs and R-loops are likely significant contributors to gene regulation. In particular, we note that NRL-interacting RNA helicases, DHX9 and DDX5, have a direct role in

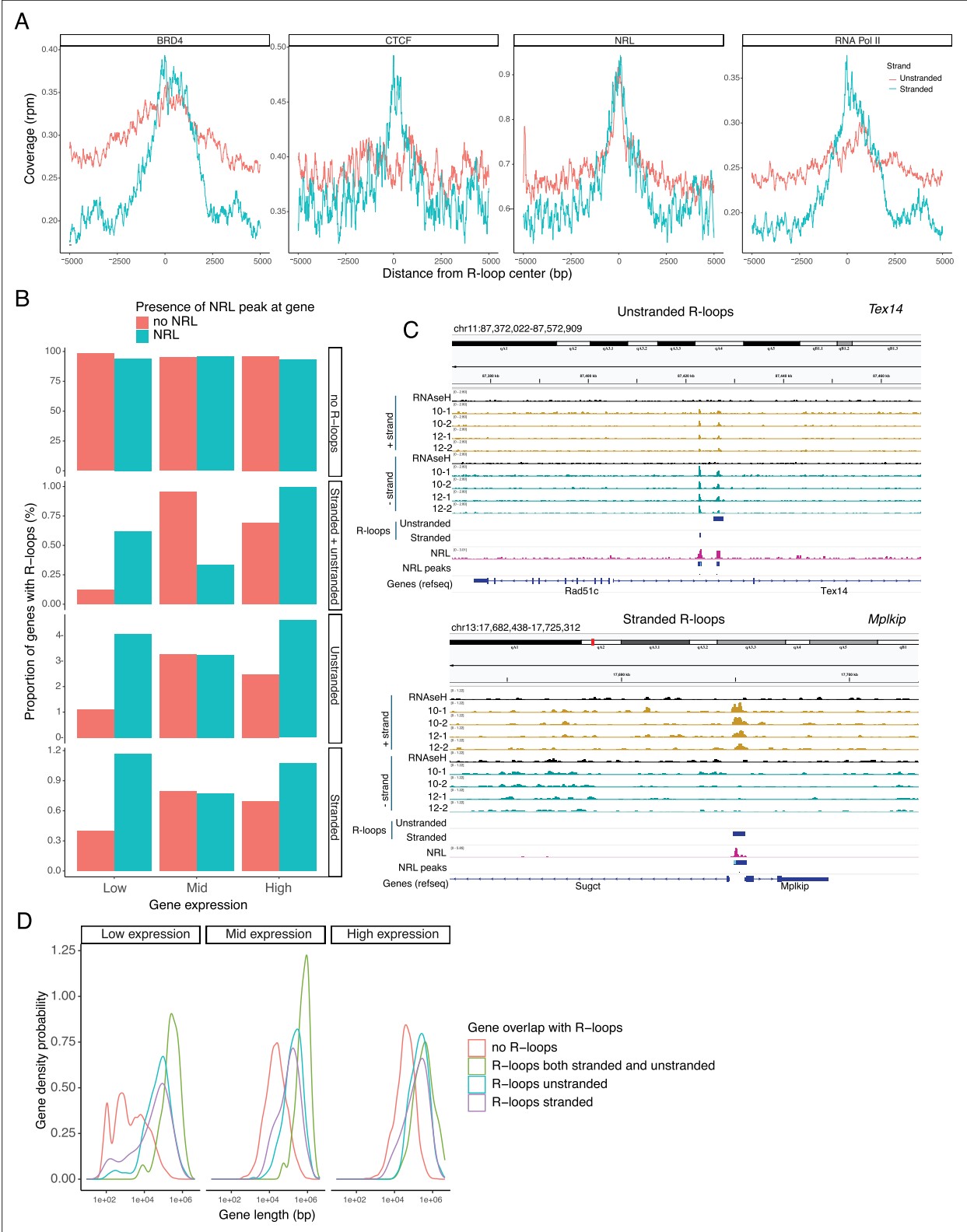

**Figure 7.** Neural retina leucine (NRL) is associated with different types of R-loops. (**A**) Metaplot of BRD4 (SRR4252658), CTCF (SRR4252685), NRL (Cut&Run), and RNA pol II (SRR4252922) signals centered on stranded and unstranded R-loop peaks. (**B**) Proportion of genes with and without stranded and unstranded R-loops and harboring NRL Cut&Run and Chip-Seq peaks. Genes were divided into three groups (low, mid, and high) based on their expression levels. (**C**) Genome view of *Tex14* and *Mplkip* mouse genes displaying ssDRIP-Seq signal in four retinas. Signals are shown for the positive

*Figure 7 continued on next page*

*Figure 7 continued*

(orange) and negative (blue) strands separately. RNase H-treated samples are pooled and shown for each strand. Peak calls for NRL and unstranded and stranded R-loops are shown in blue. (**D**) Gene density probability plot showing the distribution of R-loops over expressed genes of different lengths. Genes were divided into three groups according to their expression levels.

The online version of this article includes the following figure supplement(s) for figure 7:

**Figure supplement 1.** Distribution of neural retina leucine (NRL) peaks over genes.

**Figure supplement 2.** Absence of R-loops over the *Rho* gene.

controlling R-loop resolution and cell fate specification (*Li et al., 2020*), DNA damage response (*Saha et al., 2022*) and splicing regulation (*Chakraborty et al., 2018*). The interaction of NRL with DHX9 is very robust in GST pull downs, in HEK293 cells, and in bovine retina, suggesting that NRL–DHX9 complexes might play key roles in rod photoreceptors. Curiously, the interaction of NRL with DDX5 is weaker, suggesting that it may be more regulated or context dependent. In agreement, DHX9 but not DDX5 is enriched in retinal R-loops. Removal of RNA in bovine retina did not weaken the interaction between NRL and both DHX9 and DDX5, indicating that RNA is not required for many NRL–RBP interactions. Thus, interaction of NRL with R-loop resolvases and with R-loops shown here in vitro through R-loop pull downs, and in vivo through DRIP-Seq, suggest that NRL contributes to R-loop regulation in retinal photoreceptors. Indeed, NRL-regulated genes are enriched for stranded R-loops at promoter regions and unstranded R-loops at gene bodies implying distinct modes of R-loop-mediated gene regulation. Notably, dynamics of R-loops is proposed to regulate somatic cell reprogramming that involves interaction of SOX2 with DDX5 (*Li et al., 2020*). *Figure 8* shows a proposed model of NRL-mediated regulation of R-loops during gene expression in rod photoreceptors. Our studies thus provide an avenue to investigate the role of R-loops in retinal development and functional maintenance.

Excess of R-loops can generate DNA damage, genomic instability, and directly activate immune response (*Crossley et al., 2023*; *Niehrs and Luke, 2020*). Dysregulation of R-loops has been implicated in a variety of diseases including neurodegeneration (*Richard and Manley, 2017*). Even during normal aging, aberrant transcription contributes to mis-splicing and increased R-loop formation in *Drosophila* photoreceptors (*Jauregui-Lozano et al., 2022*). As R-loops are low or absent at many key retina genes, failure of the mechanism that maintains these genes free of R-loops could contribute to retinal disease. Interestingly, monoallelic rare *DHX9* variants increase R-loop levels and are shown to be associated with neurological disorders (*Calame et al., 2023*). Therefore, further investigation into NRL–DHX9 interaction with R-loops in the retinal photoreceptors could uncover mechanisms that maintain transcriptional homeostasis and genomic stability.

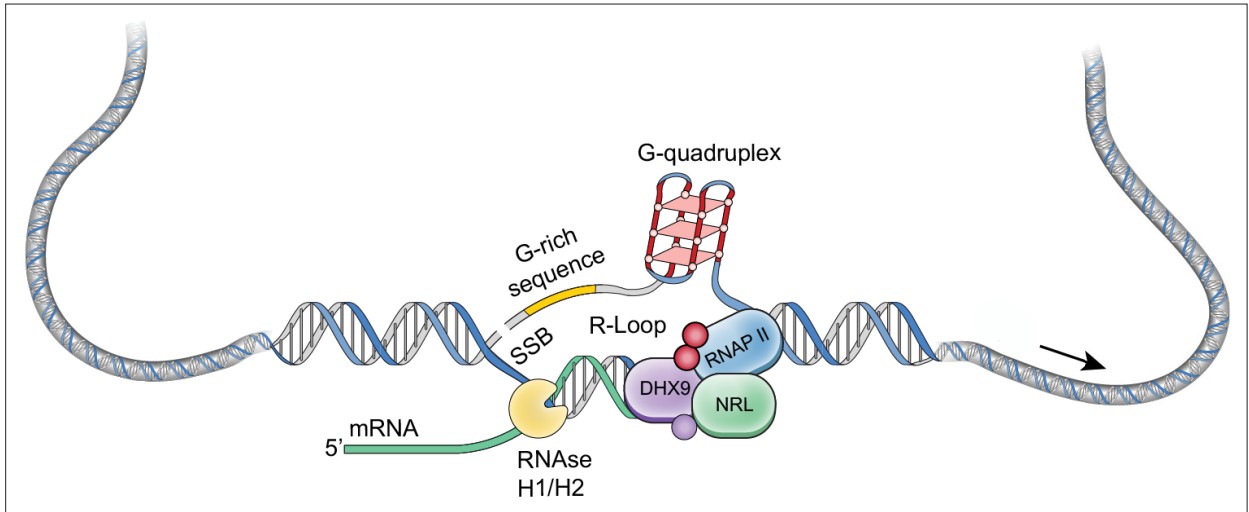

**Figure 8.** A putative model of R-loop regulation in retinal rod photoreceptors. We hypothesize that neural retina leucine (NRL) guides R-loop resolvases such as DHX9 at photoreceptor genes to critically regulate gene expression. SSB, single strand break.

Transcription factors are reported to be involved in controlling alternate splicing (*Han et al., 2017*). Based on our discovery of NRL's interaction with splicing factors and R-loop binding proteins, we hypothesize a potential role of NRL in determining patterns and rates of splicing, which is prevalent in photoreceptors (*Wan et al., 2011*; *Zelinger and Swaroop, 2018*; *Mellough et al., 2019*; *Keuthan et al., 2023*; *Ciampi et al., 2024*). NRL–RBP interactions may also influence the rate of transcription elongation and/or termination and consequently transcript isoform diversity in rod photoreceptors. Notably, mutations in splicing factors as well as aberrant splicing of many genes have been associated with retinal degeneration (*Zelinger and Swaroop, 2018*; *Liu and Zack, 2013*; *Buskin et al., 2018*).

Finally, our results indicate that other RNA-dependent chromatin processes may be influenced by NRL–RBP complexes. Many of the identified RBPs participate in maintaining chromatin architecture and modifications and in DNA damage repair, and their functions require further exploration. Some of the candidate NRL–RBP interactors are not expressed in mature rods and may bind NRL only in in vitro assays, possibly due to harboring similar domains to NRL interactors. However, this dataset is valuable because several of these proteins could be expressed during development or upregulated in disease conditions. For instance, Elavl 1 is highly expressed in mature photoreceptor cells (counts per million [CPM] = 38), providing a likely explanation for the interaction of NRL with other Elavl proteins in vitro. Notably, Elavl 2, 3, and 4 are expressed during rod development at postnatal day 6 (P6) with CPM values of 4.5, 19, and 5, respectively, and their potential role in rod photoreceptors requires further investigation. Overall, we propose that RBPs play key roles in rod photoreceptors through their interactions with NRL. Future studies will aim at unraveling the physiological significance of NRL-interacting RBPs on R-loops and chromatin regulatory mechanisms in photoreceptors.

## Methods

### Mouse strains and husbandry

All procedures involving mice were approved by the Animal Care and Use Committee of the National Eye Institute (NEI-ASP#650). C57BL/6J (B6) mice were kept in a 12-hr light/12-hr dark cycle and fed ad libitum at the NEI animal facility. Both male and female mice were used in this study.

### Human tissue

Human donor eyes were procured from Lions World Vision Institute (Tampa, FL). Autopsy material from unidentified deceased individuals is not subject to Institutional Review Board (IRB) review and does not need a determination from the Office of Human Subjects Research Protections (OHSRP) according to 45 CFR 46 and NIH policy (OHSRP ID#: 18-NEI-00619). Eyes were enucleated within 6 hr of death, followed by making a 2-cm incision at the limbus before immersing in a solution of 4% formaldehyde in PBS. Eyes were subsequently transferred to PBS for shipment and kept at 4°C. Before sectioning, eyes were fixed in 4% formaldehyde in PBS for 2 hr and immersed in 30% sucrose.

### Antibodies

The following antibodies were used: DDX5 Monoclonal antibody (Cat. No. 67025-1-Ig), DHX9 Polyclonal antibody (Cat. No. 17721-1-AP), and HNRNPA2B1 (Cat. No. 14813-1-AP), all obtained from Proteintech (Rosemont, IL, USA); HNRNPA1 (Cat.No. 8443, Cell Signaling, Danvers, MA, USA); DNA–RNA Hybrid antibody S9.6 (Cat. No. 65683, Active Motif, Carlsbad, CA, USA); Xpress Monoclonal Antibody (Cat. No. R910-25, Thermo Fisher, Waltham, MA, USA); HNRNPM (Cat. No. A6937, Abclonal, Woburn, MA, USA); HNRNPU and HNRNPUl1 (Cat. No. MA1-24632 and Cat. No. 10578-1-AP, respectively, Thermo Fisher, Waltham, MA); rabbit anti-NRL (Swaroop lab (*Liang et al., 2022*) and R&D systems Cat. AF2945); double-stranded DNA (Cat. No. MAB030, MilliporeSigma, Burlington, MA, USA).

### Plasmids

Human NRL constructs in pcDNA4/HisMax vector containing an N-terminal Xpress tag (Cat. No. V86420, Thermo Fisher Scientific, Waltham, MA, USA) or in pGEX-4 T2 plasmid containing an N-terminal GST tag (GE Healthcare/Cytiva; cytivalifesciences.com) have been described previously (*Liang et al., 2023*). Human RNase H1 or D201N catalytic dead mutant EGFP fusions (GFP-HR and GFP-dHR)

were developed in the Cimprich lab (*Crossley et al., 2023*) and obtained from Addgene (Cat. No. 196702 and 196703).

## Mass spectrometry

Affinity-purified protein samples after incubation of GST-NRL (*n* = 3 for GST-NRL; *n* = 3 for GST only control) with bovine (*n* = 2 for NRL immunoprecipitation, IP; *n* = 1 for IgG control) or mouse retina (*n* = 2 for NRL IP, *n* = 1 for IgG control) were subjected to liquid chromatography by tandem mass spectrometry (LC–MS–MS) analysis (Poochon Scientific, Frederick, MD, USA). Samples were digested with trypsin, peptides were extracted, desalted and analyzed using Q-Exactive hybrid Quadrupole-Orbitrap Mass Spectrometer and Thermo Dionex UltiMate 3000 RSLCnano System (Thermo Fisher Scientific, Waltham, MA, USA). Peptides were ionized and sprayed into the mass spectrometer using Nanospray Flex Ion Source ES071 (Thermo Scientific, Waltham, MA, USA). Raw data files were searched against bovine, mouse or human protein sequence databases (National Center for Biomedical Information, NCBI) using Proteome Discoverer 1.4 software (Thermo Scientific, Waltham, MA, USA) based on SEQUEST algorithm. Carbamidomethylation of cysteines was set as a fixed modification, whereas oxidation and deamidation Q/N-deamidated (+0.98402 Da) were set as dynamic modifications. The minimum peptide length was specified to be five amino acids. The precursor mass tolerance was set to 15 ppm and fragment mass tolerance to 0.05 Da. The maximum false peptide discovery rate was specified as 0.01. All assembled proteins with peptides sequences and matched spectrum counts (PSM counts [#PSM]) were included in the analysis.

## Y2H screening

We performed Y2H assays using a modified Matchmaker System (Clonetech, Mountain View, CA, USA) with a human retinal cDNA prey library pGADT7 vector (containing Gal4 activating Domain). Two partial NRL domains were synthesized by Genewiz (Germatown, MD, USA): (1) NRL EHD with the basic motif (BM) (residues 95–175), and (2) partial NRL BM with NRL bZIP domain (residues 176–237). NRL domain containing inserts were subcloned into the bait vector, pGBKT7 (containing Gal4-binding domain), and transformed into Y2HGold yeast. The yeast containing a bait were inoculated at 30°C overnight in 50 ml (−)Trp broth to obtain a cell yield of 7.5 × 10$^8$ cells per culture. Subsequently, yeast cells were pelleted at 3000 × *g* for 5 min, resuspended in 50 ml of (−)Trp broth and incubated at 30°C for 3–4 hr with shaking (225 rpm). Yeast colonies containing baits were transformed with 10 µg of human retinal prey cDNA library. For screening, positive colonies were selected on SD/-Trp/-Leu/X-alpha-gal/Aureobasidin-A (DDO/X/A) media to determine β-galactosidase activity and antibiotic resistance as an indicator of a positive interaction with NRL. These interactors were validated by patch plating onto higher stringency SD/-Trp/-Leu/-Ade/-His/X-alpha-gal/Aureobasidin-A (QDO/X/A) media. Positive interactor inserts were PCR amplified, sequenced using T7 primer, and identified by Blastn and Blastx (NCBI).

To confirm NRL-interacting RBPs, colonies containing RBP prey and NRL domain baits were resuspended in ultra-pure water, and plated against controls on five different plating media: SD/-Trp, SD/-Leu, SD/-Trp/-Leu, SD/-Trp/-Leu/X-alpha-gal/Aureobasidin-A, and SD/-Trp/-Leu/-Ade/-His/X-alpha-gal/Aureobasidin-A.

## Cell culture and transfection

HEK293 cells were obtained from American Type Culture Collection (ATTC) (Cat. No. CRL-1573, Manassas, VA, USA), authenticated with STR profiling and tested negative for Mycoplasma. Cells were cultured in Dulbecco's modified Eagles's medium (Cat. No. 11885084, Thermo Fisher Scientific, Grand Island, NY, USA) containing 10% fetal calf serum (Cat. No. S11550, R&D Systems, Flowery Branch, GA, USA), 100-units/ml penicillin G and 100 µg/ml streptomycin (Cat. No. 15140122, Thermo Fisher Scientific, Grand Island, NY, USA). Cell transfection was performed with X-tremeGENE 9 DNA Transfection Reagent (Cat. No. 6365779001, Cat. No. 06365787001, Roche, Mannheim, Germany) or Lipofectamine 2000 (Cat. No. 11668027, Thermo Fisher Scientific, V Carlsbad, CA, USA) per the manufacturer's instructions.

## Immunofluorescence and microscopy

Eyes from *Nrl* WT or *Nrl*-knockout (KO) C57BL/6 mice (*Mears et al., 2001*) at postnatal day (P) 28 were fixed in 4% paraformaldehyde for 1 hr, washed 3× in PBS, and embedded in optimal cutting temperature (OCT) medium on dry ice. For NRL detection and PLA, eyes were directly embedded in OCT. Human eyes were fixed for 2 hr in 4% PFA, washed as above and embedded in OCT. Eyes were cryosectioned at 14 µm and mounted on SuperFrost Plus slides (Thermo Fisher Scientific, Waltham, MA, USA). For NRL detection and PLA in mouse retinas, fresh sections were postfixed in 4% PFA for 7 min. Retinal sections were then incubated in blocking solution (5% bovine serum albumin [BSA], 0.3% Triton X-100 in PBS) for 1 hr followed by primary antibody (anti-DDX5, anti-DHX9, or anti-NRL 1:100) incubation in blocking solution at 4°C overnight. After three washes of 10 min each in PBS, sections were incubated in secondary antibody in blocking solution in the presence of the DNA dye DAPI (4',6-diamidino-2-phenylindole) for 1 hr. Sections were washed 3× in PBS and mounted in ProLong Gold Antifade Reagent (Life Technologies Inc, Carlsbad, CA, USA). Confocal images were acquired using SP8 Leica 2-photon confocal microscope or a Zeiss LSM 800 point scanning confocal microscope using the AiryScan detector.

## Proximity ligation assay

PLA was carried out using Duolink PLA Fluorescence kit (Millipore, Sigma) per the manufacturer's instructions. Briefly, HEK293 cells were transfected with NRL-Xpress or RNase H1 constructs (GFP-HR and GFP-dHR). Cells were washed with PBS 48 hr post-transfection, fixed using 4% paraformaldehyde in PBS for 15 min at room temperature and washed three times with PBS. Mouse and human retinas were fixed as described above. Cells or sections were permeabilized using 0.3% Triton X in PBS for 5 min. Duolink blocking solution was added for 60 min at 37°C. Rabbit anti-DHX9, anti-DDX5 antibodies, anti-NRL, or Xpress antibodies were diluted in Duolink antibody diluent and incubated overnight at 4°C. After washing, plus and minus PLA probes were diluted 1:5 into Duolink antibody diluent. Cells were incubated at 37°C for 1 hr. After further washing, the ligation was performed for 30 min at 37°C.

## Protein lysates and enzymatic treatments

### Bovine Sephacryl S-300 fractions

High molecular mass complexes containing NRL were obtained by size exclusion chromatography as described earlier (*Liang et al., 2023*). Briefly, bovine retinal nuclear extracts were fractionated using 40% ammonium sulfate. Isolation of high molecular mass protein complexes was performed using HiPrep 16/60 Sephacryl S-300 High-Resolution column (GE Lifesciences/Cytiva). The peaks corresponding to 650 and 450 kDa containing NRL were immunoprecipitated using NRL antibody/protein A bead complexes overnight at 4°C, as described earlier.

### Bovine nuclear extracts for GST purification

Retinas were resuspended in CE1 buffer (10 mM HEPES-KOH pH 7.5, 1.5 mM $MgCl_2$, 10 mM KCl, 10% glycerol, 1 mM DTT) with protease inhibitors (PI) for 10 min at 4°C. Retinas were homogenized in dounce homogenizer 25 times and pelleted at 250 × *g* for 10 min. Nuclear pellets were resuspended in high salt buffer (450 mM NaCl, 10 mM Tris pH 7.4, 2 mM EGTA, 0.1% Triton X-100, 2 mM $MgCl_2$) with PI for 30 min at 4°C. Chromatin was pelleted at 10,000 × *g* for 10 min. Supernatants were diluted to 150 mM NaCl and incubated with BSA-blocked antibody–bead conjugates overnight as described earlier. For RNase A treatments, lysates were incubated with RNase A at 1 µg per µg DNA for 30 min at 37°C before antibody incubation.

### Bovine and HEK293 nuclear extracts and treatments

HEK293 cells cultured in 6-well plates as described earlier, were lysed in 1 ml co-IP binding buffer (40 mM Tris-HCl pH 8.0, 150 mM NaCl, 2 mM EDTA, and 0.2% NP-40) containing PI cocktail. The nuclear pellet from bovine retina was obtained as described above and resuspended in 1 ml of co-IP binding buffer. After sonication using an ultrasonic liquid processor (MISONIX, Inc, Farmingdale, NY) until complete chromatin dissolution, samples were clarified by centrifugation at 16,000 × *g* for 10 min at 4°C. The DNA concentration in the lysate was measured by Qubit dsDNA BR Assay Kit (Thermo

Fisher Scientific, Waltham, MA). For each immunoprecipitation, 100 µl lysate was treated as follows: For control, the lysate was mixed with 50 units of RNaseOUT recombinant ribonuclease inhibitor (Thermo Fisher Scientific, Waltham, MA, USA); for RNAse A treatment, the lysate was mixed with 5 µg of RNase A per µg of DNA (Thermo Fisher Scientific, Waltham, MA, USA); for RNase H treatment, 2.5 units of RNase H were added per µg of DNA with 1× RNase H buffer (New England Biolabs, Ipswich, MA); for DNase I treatment, 1 unit of DNase I was added per µg of DNA with 1× DNase I reaction buffer (Thermo Fisher Scientific, Waltham, MA). All reactions were performed for 30 min at 37°C. After treatments, the lysates were incubated with antibodies as described above.

## Co-immunoprecipitation

Dynabeads Protein A (Thermo Fisher Scientific, Waltham, MA) were blocked with 1% BSA at room temperature for 1 hr and incubated with 1 µg of the respective antibodies for another hour. After washing off the unbound antibody, the lysates were added to the beads and incubated overnight at 4°C. The beads were then washed three times for 15 min each with co-IP wash buffer (150 mM NaCl, 10 mM Tris-HCl pH 7.4, 2 mM EGTA, 1% Triton X-100, 2 mM MgCl$_2$) and bound protein complexes were eluted by boiling in Laemmli SDS sample buffer for 10 min at 95°C.

## GST affinity purification

GST-NRL was expressed in BL21(DE3) competent cells (Thermo Fisher Scientific, Waltham, MA) using 0.2 mM Isopropyl ß-D-1-thiogalactopyranoside (IPTG) for 16 hr at 20°C and purified using Glutathione Sepharose beads (Cat. No. GE17-0756-01, MilliporeSigma, Burlington, MA) by incubating BSA-blocked beads with cell lysates for 3 hr at 4°C. GST-NRL beads were then incubated for 2 hr at 4°C with nuclear fraction from bovine retina (prepared as above) with or without 25 U of benzonase. Afterwards, the beads were washed three times for 10 min each in PBS with 1% Triton X-100, then boiled in Laemmli sample buffer containing 100 mM DTT for 7 min and stored at –80°C or loaded into 4–15% acrylamide gels for immunoblotting.

## R-loop detection

R-loops were extracted as previously reported (*Sanz and Chédin, 2019*) with some modifications. Retinas or HEK293 cells were incubated in nuclear extraction buffer (Tris pH 7.4 10 mM, NaCl 10 mM, MgCl$_2$ 3 mM, BSA 1%, Tween-20 0.1 %, NP-40 0.1%) for 10 min. After dounce homogenization, nuclei were pelleted at 500 × *g* for 5 min at 4°C. Genomic DNA was obtained by incubating nuclei in 10 mM Tris-HCl containing 1 mM EDTA (TE) with 1% SDS and 60 µg Proteinase K20 overnight at 37°C. DNA was extracted with phenol/chloroform isoamyl alcohol 25:24:1 (PCI). Briefly, equal volume of PCI was added to each sample. After gentle mixing, samples were spun down at 12,000 × *g* for 30 min at 4°C. DNA was precipitated by adding 1/10 volume of 3 M NaOAc, pH 5.2 and 2.5 volumes of 100% (vol/vol) ethanol. The pellet was washed with 1 ml of 70% ethanol. After gentle mixing, samples were centrifuged at 12,000 × *g* for 15 min at 4°C. The supernatant was discarded, and the pellet was air dried and resuspended in TE buffer. DNA was incubated with RNase H buffer and RNase III at 37°C OV with and without RNase H.

To detect R-loops during development, retinas were collected from three male B6 mice at each time point (postnatal day (P)6, P10, P14, and P28).

### Dot-blot analysis

gDNA samples diluted in TE buffer were blotted on a positively charged Hybond -N+hybridization membrane (Thermo Fisher Scientific, Waltham, MA, USA) using a dot-blot apparatus (Cat. No. 1706545, Bio-Rad Laboratories, Hercules, CA, USA). Membranes were crosslinked using 1200 µJ UV light.

## Electrophoretic mobility shift assay

DNA oligonucleotides (5 pmol, forward strand) were incubated with 50 µCi of $^{32}$P-ATP g and 10 units of T4 polynucleotide kinase for 30 min at 37°C. The reaction was terminated by heating at 65°C for 10 min and column purified to remove unincorporated label. This labeled oligonucleotide was annealed to 5 pmol of the complementary strand by heating to 95°C for 5 min and cooling slowly to room temperature. Double-stranded $^{32}$P-labeled oligonucleotides (0.2 pmol, 20,000 cpm) were

incubated with 5 µg of bovine nuclear and 1× binding buffer (LightShift, Thermo Fisher Scientific), 50 ng/µl Poly dI-dC, and 5 mm $MgCl_2$ for 1 hr. Reactions were terminated by addition of loading buffer, separated on 8% DNA retardation gel (Invitrogen, Waltham, MA, USA). Subsequently, the gel was dried and exposed to Phosphor screen overnight, which was scanned using Typhoon FLA 9500 Biomolecular Imager (GE Lifesciences). Cold competition assays were performed by preincubating unlabeled double-stranded oligonucleotides at 0.1–0.2 pmol concentration, as indicated, with the nuclear extract for 5 min before the addition of the labeled probes. Mutant probes were incubated at 0.2 pmol.

Probes for EMSA were designed by examining putative NRL-binding motifs in ATAC-Seq footprints overlapping with NRL ChIP-Seq peaks as described *Marchal et al., 2022*.

Probe for *DHX9*: 5′ GAGGTTGCTGAGCCCCGCCCCCTC 3′.
Mutant *DHX9* probe 1: 5′ GAGGTCAATTTGCCCCGCATTCTC 3′.
Mutant *DHX9* probe 2: 5′ GAGGTACCCTTGCCCTTCAACCTC 3′.

## Protein network

RBPs included in the analysis had >twofold increase compared to the controls and were present at 10 or more total PSMs in experimental samples. PPIs among RBPs were extracted from STRING Database (https://string-db.org/). Only data identified experimentally were included for analysis. The confidence threshold was set to 0.4 and displayed in the network as edge thickness. Networks were created using Cytoscape (https://cytoscape.org/).

## Gene regulation analysis

Normalized transcripts per million of neuronal cells from single-cell human data (https://www.protein-atlas.org/) was used to compare expression levels in photoreceptors and other neuronal cells. Over-laps between genes encoding RBPs (±1 kb from gene body) identified in two out of four assays with human retina NRL-ChIP-Seq peaks and super enhancers from human retina (obtained from *Marchal et al., 2022*) were identified using bedtools (https://bedtools.readthedocs.io/en/latest/). Genomic views of gene loci were obtained using IGV (https://software.broadinstitute.org/software/igv/).

## S9.6 DNA–RNA co-IP

DRIP was performed as described (*Li et al., 2020*; *Ginno et al., 2012*), with minor modifications. Genomic DNA was digested with 50 U of Mse I, Dde I, Alu I, and Mbo I. Digested DNA (500 ng) was treated with RNase III and/or RNase H overnight as above. gDNA was added to Protein G Dynabeads (Thermo Scientific) preincubated with 1.5 µg of S9.6 antibody for 1 hr at RT in 500 µl of binding buffer (10 mM $Na_2HPO_4$, 140 mM NaCl, 0.05% Triton X-100). After three washes in binding buffer, R-loop-antibody–protein G complex was incubated with mouse retina lysate (200 µg) overnight at 4°C. Nuclear lysates were obtained as described above and pretreated with RNase A (0.1 ng per 1 ug DNA) for 1 hr at 37°C followed by incubation with 1000 U RNaseOut for 10 min at 4°C. After three washes with IP buffer, S9.6-bound complexes were analyzed by immunoblotting.

## ssDRIP-Seq

ssDRIP-Seq was performed as described earlier (*Li et al., 2020*). The libraries were prepared using the Accel-NGS 1S Plus DNA Library Kit (IDT technologies Coralville, IA, USA). Briefly, the DRIPed DNA sample obtained as above was fragmented to an average length of 250 base pairs (bp) utilizing an S220 Focused-ultrasonicator (Covaris, Woburn, MA, USA). The fragmented DNA was then denatured by heating at 98°C for 2 min, followed by immediate cooling on ice for 2 min. The R2 adapter was first ligated to the 3′ end of the ssDNA, followed by an extension step. Subsequently, the R1 adapter was ligated to the 5′ end. After PCR amplification, the resulting libraries were purified using AMPure XP beads (Beckman Coulter, San Jose, CA). The quality of each library was assessed with a TapeStation instrument (Agilent, Palo Alto, CA, USA), and sequencing was carried out on an Illumina NextSeq 2000 system (San Diego, CA, USA) at 2 × 100 bp at a final library concentration of 1100pM with a sequencing dept of grater that 20 million pair end reads.

After trimming adaptor sequences, low-complexity tails were removed by further trimming 10 bps from each read on the 3′ end of the first read in the pair and 5′ end of the second read in the pair. Reads

were aligned to the mouse genome (mm10) using Bowtie2. Duplicated reads, reads with score <20 (including multi-mapping reads) and reads mapping ENCODE consortium blacklisted regions of mm10 genome (https://github.com/Boyle-Lab/Blacklist/; *Boyle, 2021*) were removed. Mapped reads were split by strand using samtools. To confirm the quality of the experiment, a PCA was run on the 500 bp coverage scaled in R for all samples treated or not with RNAse-H, using R ggfortify package. R-loops calling was performed using MACS2. First, all R-loops (without regard of any strand specificity) were called on all reads, using the merged RNase H-treated samples as control dataset using MACS2 with the default parameters (narrow peaks) and the broad peaks option. R-loops found in at least three samples with a *q*-value ≤0.001 in the narrow call or in the broad call were merged. From this list of R-loops, strand specificity was assessed for each R-loop. To do so, peaks presenting signal enrichment from one strand over the other were called on reads split by strand, using the reads mapping the opposite strand as control dataset, similarly to the R-loops call. By intersecting the peaks presenting signal enrichment from one strand over the other and the total R-loops, we extracted the strand-specific R-loops and the unstranded R-loops. The annotation of R-loops, gene ontology analysis, and chromatin marks enrichment were computed using HOMER and plotted using R package ggplot2. Overlap between gene expression levels, R-loops and NRL peaks were computed using Bedtools. For this, we utilized our previously published expression data that were reanalyzed for GRCm38/ENSv98 annotation (*Brooks et al., 2019*), re-analyzed NRL chromatin-binding data (ChIP-Seq from *Hao et al., 2012* and Cut&Run from *Liang et al., 2022*). Overlap between R-loops and NRL target genes (*Liang et al., 2022*; *Supplementary file 5*) was computed using Bedtools. The gene promoter-TSS region was defined as –2.5 to +0.5 kb from the TSS. The gene downstream region was defined as –0.5 to 2.5 kb from the gene end. We generated related plots using the R package ggplot2 and chromatin profiles using IGV.

## Statistical analysis

Means between groups were compared using Student's *t*-test in at least three biological replicates. The differences were considered statistically significant with a two- or one-tailed p-value of <0.05. Immunoblots of pull downs were analyzed using ImageJ (https://imagej.net/ij/) by calculating the ratio of signal intensities of prey and bait normalized to controls. Dot-blot signal intensities were normalized to input gDNA. R-loop protein enrichment was calculated using the Fisher's exact test with a contingency table based on the total number of retina-expressed nuclear RBPs. Mouse nuclear RBPs data were obtained from the RBP2GO database (https://rbp2go.dkfz.de/), which reports 1764 RBPs including 1534 that are expressed in the mouse retina.

## Acknowledgements

The authors are grateful to James Friedman for the human retina 'prey' cDNA library, Jindan Yu and James Friedman for some of the early observations on DDX5, and Kenneth P Mitton for advice on Y2H experiments. We also thank Robert Fariss (NEI Imaging Core) for assistance with confocal microscopy, Vishal Dandewad for some of the R-loop experiments, Anupam Mondal for PPI networks, Suja Hiriyanna with cloning, and Zachary Batz for GEO submission. This research is supported by Intramural Research Program of the National Eye Institute (Z01EY000450 and Z01EY000546) to AS and K99/R00 award (1K99EY030918-01) to XCD.

## Additional information

### Competing interests

Claire Marchal: director and founder of In silichrom Ltd. The other authors declare that no competing interests exist.

### Funding

| Funder | Grant reference number | Author |
|---|---|---|
| National Eye Institute | Z01EY000450 | Anand Swaroop |

| Funder | Grant reference number | Author |
|---|---|---|
| National Eye Institute | Z01EY000546 | Anand Swaroop |
| National Eye Institute | K99EY030918 | Ximena Corso Diaz |

The funders had no role in study design, data collection and interpretation, or the decision to submit the work for publication.

## Author contributions

Ximena Corso Diaz, Conceptualization, Formal analysis, Funding acquisition, Validation, Investigation, Methodology, Writing – original draft, Writing – review and editing; Xulong Liang, Formal analysis, Investigation, Methodology, Writing – review and editing; Kiam Preston, Validation, Investigation, Methodology, Writing – review and editing; Bilguun Tegshee, Jacob Nellissery, Sharda Prasad Yadav, Investigation, Methodology, Writing – review and editing; Milton A English, Resources, Investigation, Methodology, Writing – review and editing; Claire Marchal, Data curation, Software, Formal analysis, Validation, Visualization, Writing – original draft, Writing – review and editing; Anand Swaroop, Conceptualization, Resources, Formal analysis, Supervision, Funding acquisition, Writing – original draft, Project administration, Writing – review and editing

## Author ORCIDs

Ximena Corso Diaz (iD) https://orcid.org/0009-0001-0230-7757
Anand Swaroop (iD) https://orcid.org/0000-0002-1975-1141

## Ethics

All procedures involving mice were approved by the Animal Care and Use Committee of the National Eye Institute (NEI-ASP#650). C57BL/6J (B6) mice were kept in a 12-hr light/12-hr dark cycle and fed ad libitum at the NEI animal facility. Both male and female mice were used in this study.

Reviewer #1 (Public review): https://doi.org/10.7554/eLife.103259.3.sa1
Reviewer #2 (Public review): https://doi.org/10.7554/eLife.103259.3.sa2
Author response https://doi.org/10.7554/eLife.103259.3.sa3

# Additional files

## Supplementary files

Supplementary file 1. Neural retina leucine (NRL)-interacting retinal proteins identified by liquid chromatography with tandem mass spectrometry (LC–MS–MS) after purification with NRL-GST. Peptide spectrum match (PSM) values are shown for each protein. The listed proteins were enriched in three independent experiments.

Supplementary file 2. Neural retina leucine (NRL)-interacting retinal proteins identified by liquid chromatography with tandem mass spectrometry (LC–MS–MS) from bovine retinal fractions and mouse retina. RNA-binding proteins (RBPs), all mouse proteins, and all bovine proteins are shown in different tabs.

Supplementary file 3. R-loop-interacting proteins from this study and published datasets.

Supplementary file 4. Genes from the RetNet database harboring R-loops. Genes with neural retina leucine (NRL) peaks are depicted as yes (Y).

Supplementary file 5. Neural retina leucine (NRL)-regulated genes at different mouse postnatal (P) time points.

MDAR checklist

## Data availability

All sequencing data that support the findings of this study have been deposited in the National Center for Biotechnology Information Gene Expression Omnibus (GEO) and are accessible through the GEO accession number GSE274666.

The following dataset was generated:

| Author(s) | Year | Dataset title | Dataset URL | Database and Identifier |
|---|---|---|---|---|
| Corso-Díaz X | 2025 | Interaction of bZIP transcription factor NRL with RNA-binding proteins implicates R-loops in photoreceptor gene regulation | https://www.ncbi.nlm.nih.gov/geo/query/acc.cgi?acc=GSE274666 | NCBI Gene Expression Omnibus, GSE274666 |

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
