## [Editor Report · eLife Assessment]

This **important** study employed multiple orthogonal techniques and tissue samples to investigate the interaction between the NRL transcription factor and RNA-binding proteins in the retina. The findings are **convincing** to support an interaction between NRL and the DHX9 helicase. The significance of the study could be enhanced with functional experiments of NRL-R-loop interactions in the developing retina and their potential role in photoreceptor health and gene regulation.

---

## [Referee Report · Reviewer #1 (Public review)]

Summary:

In this manuscript, Corso-Diaz et al, focus on the NRL transcription factor (TF), which is critical for retinal rod photoreceptor development and function. The authors profile NRL's protein interactome, revealing several RNA-binding proteins (RBPs) among its components. Notably, many of these RBPs are associated with R-loop biology, including DHX9 helicase, which is the primary focus of this study. R-loops are three-stranded nucleic acid structures that frequently form during transcription. The authors demonstrate that R-loop levels increase during photoreceptor maturation and establish an interaction between NRL TF and DHX9 helicase. The association between NRL and RBPs like DHX9 suggests a cooperative regulation of gene expression in a cell-type-specific manner, an intriguing discovery relevant to photoreceptor health. Since DHX9 is a key regulator of R-loop homeostasis, the study proposes a potential mechanism where a cell-type-specific TF controls the expression of certain genes by modulating R-loop homeostasis. The authors also identify another R-loop resolvase, DDX5 as having weaker interaction with NRL, perhaps due to indirect mechanism.

This is a very interesting study providing genome-wide mapping of R-loops in mammalian retina, which shows an enrichment of R-loops over intergenic regions as well as genes encoding neuronal function factors. The R-loop-enriched genes are longer than genes without R-loops, which supports previous findings from studies in neuronal cells. This is a very relevant study highlighting the possible mechanism of gene expression regulation via interactions between TFs, RNA binding proteins, and R-loops. In that regard, it would be very interesting to uncover the biological relevance of such gene regulation. The authors provide adequate evidence of interaction between R-loops and NRL TF via in vitro IP assay and genomic co-localization, however, this interaction can be mediated via multiple R-loop or RNA-binding proteins. Thus, follow-up studies would be appropriate to characterize this interaction in more detail.

---

## [Referee Report · Reviewer #2 (Public review)]

Summary:

The Authors utilize biochemical approaches to determine and validate NRL protein-protein interactions to further understand the mechanisms by which the NRL transcription factor controls rod photoreceptor gene regulatory networks. Observations that NRL displays numerous protein-protein interactions with RNA-binding proteins, many of which are involved in R-loop biology, led the authors to investigate the role of RNA and R-loops in mediating protein-protein interactions and profile the co-localization of R-loops with NRL genomic occupancy.

Strengths:

Overall, the manuscript is well-written, providing succinct explanations of the observed results and potential implications. Additionally, the Authors use multiple orthogonal techniques and tissue samples to reproduce and validate that NRL interacts with DHX9 and DDX5. Experiments also utilize specific assays to understand the influence of RNA and R-loops on protein-protein interactions. The Authors also use state-of-the-art techniques to profile R-loop localization within the retina and integrate multiple previously established datasets to correlate R-loop presence with transcription factor binding and chromatin marks an attempt to understand the significance of R-loops in the retina.

Weaknesses:

In general, the Authors provide interpretations of the data that fit a narrative about NRL and the perceived significance of interactions with RNA binding proteins. Large-scale screens for NRL protein interactions were conducted but all of the data is not reported. For example, NRL IP-Mass Spec was performed, but the authors only provide interaction/detection data for identified interactions with known RNA binding proteins. We cannot assess the enrichment of interactions or specificity of interactions with RNA binding proteins based on the reported results. Additionally, the lack of experiments testing the functional significance of Nrl interactions with R-loops within the developing retina fails to provide novel biological insights into the regulation of gene regulatory networks. While this provides additional avenues for research in the future, it is unclear that NRL interaction with R-loops have physiological relevance for photoreceptor health or function.

---

## [Author Response]

The following is the authors’ response to the original reviews.

**Public Reviews:**

**Reviewer #1 (Public review):**
Summary:In this manuscript, Corso-Diaz et al, focus on the NRL transcription factor (TF), which is critical for retinal rod photoreceptor development and function. The authors profile NRL's protein interactome, revealing several RNA-binding proteins (RBPs) among its components. Notably, many of these RBPs are associated with R-loop biology, including DHX9 helicase, which is the primary focus of this study. R-loops are three-stranded nucleic acid structures that frequently form during transcription. The authors demonstrate that R-loop levels increase during photoreceptor maturation and establish an interaction between NRL TF and DHX9 helicase. The association between NRL and RBPs like DHX9 suggests a cooperative regulation of gene expression in a cell-type-specific manner, an intriguing discovery relevant to photoreceptor health. Since DHX9 is a key regulator of R-loop homeostasis, the study proposes a potential mechanism where a cell-type-specific TF controls the expression of certain genes by modulating R-loop homeostasis. This study also presents the first data on R-loop mapping in mammalian retinas and shows the enrichment of R-loops over intergenic regions as well as genes encoding neuronal function factors. While the research topic is very important, there is some concern regarding the data presented: there are substantial data supporting the interaction between NRL and DHX9, including pull-down experiments and proximity labeling assay (PLA), however, the data showing an interaction between NRL and DDX5, another R-loop-associated helicase, are inadequate. Importantly, the data supporting the claim that NRL interacts with R-loops are absolutely insufficient and at best, correlative. The next concerns are regarding the R-loop mapping data analysis and visualization.Strengths:There is compelling evidence that the NRL transcription factor interacts with several RNA binding proteins, and specifically, sufficient data supporting the interaction of NRL with DHX9 helicase.A major strength is the use of the single-stranded R-loop mapping method in the mouse retina.Weaknesses:(1) Figure S1A: There is a strong band in GST-IP (control IP) for either HNRNPUI1 or HNRNPU, although the authors state in their results that there is a strong interaction of these two RBPs with NRL.

Under our experimental conditions, most RNA-binding proteins displayed higher binding to glutathione beads (Fig. S1A). However, GST-NRL purifications showed much stronger signals for respective RBPs. In the case of HNRNPU and HNRNPUl1, white bands that are indicative of substrate depletion due to higher protein levels are observed in GST-NRL lanes. Additionally, in Figures 1B and 1C, there is a clear enrichment of HNRNPU and HNRNPUl1 above the background signal. We added this to the text. See page 5.

Both DHX9 and DDX5 samples have a faint band in the GST-IP.

RNA-binding proteins may display some background as observed in other studies (e.g. PMID: 32704541). We think that showing the raw data without decreasing the exposure time is useful and that there is a clear enrichment compared to controls. In addition, we tested the interaction in multiple systems.

There is an extremely faint band for HNRNPA2B1 in the GST-NRL IP lane. Given this is a pull-down with added benzonase treatment to remove all nucleic acids, these data suggest, that previously observed NRL interactions with these particular RBPs are mediated via nucleic acids. Similarly, there is a loss of band signal for HNRNM in this assay, although it was identified as an NRL-interacting protein in three assays, which again suggests that nucleic acids mediate the interaction.

Thank you for highlighting this point. We mention in the manuscript that the interaction between HNRNPM and A1 depends on nucleic acids, as noted by the reviewer, since there is no obvious band after the pull-down. We have now added that the interaction of NRL with HNRNPA1B1 is likely dependent on nucleic acids as well, given its weak signal. See page 5.

(2) The data supporting NRL-DDX5 interaction in rod photoreceptor nuclei is very weak. In Figure 2D, the PLA signal for DDX5-NRL is very weak in the adult mouse retina and is absent in the human retina, as shown in Figure 2H.

We agree with the reviewer. We think that the signal for DDX5 is weak, and we addressed this in the text. We noted on page 7: “Taken together, these findings suggest a strong interaction between NRL and DHX9 throughout the nuclear compartment in the retina and that a transient and/or more regulated interaction of NRL with DDX5 may require additional protein partners.” We have modified this sentence to add that the data also suggest transient interaction or the requirement of additional protein partners for stable interaction. See page 7.

Given that there is no NRL-KO available for the human PLA assay, the control experiments using single-protein antibodies should be included in the assay. Similarly, the single-protein antibody control PLA experiments should be included in the experimental data presented in Figure 2J.

Thank you for the suggestion. We performed PLAs using both DHX9 and IgG in the human retina and observed no specific amplification signal. Some background is observed outside the nucleus and in the extracellular space. We added these results to the text and to the supplementary information. See page 7 and Fig.S2B.

(3) The EMSA experiment using a probe containing NRL binding motif within the DHX9 promoter should include incubation with retina nuclear extracts depleted for NRL as a control.

In EMSA experiments, we used bovine retina to obtain enough protein quantities. As suggested by the reviewer, using NRL depleted extract would increase the specificity of observed gel shift and complement our pre-immune serum as a negative control. However, removal of all the NRL protein using the antibodies available was not feasible. In the future, we will use enough mice to obtain large quantities of protein for this experiment and will collect retinas from Nrl knockout as negative control.

(4) There is a reduced amount of DHX9 pulled down in NRL-IP in HEK293 cells, but there is no statistically significant difference in the reciprocal IP (DHX9-IP and blotting for NRL) (Figure 4C).

We believe the reviewer is referring to the data in Figure 4C showing that RNase H treatment led to significantly reduced pulldown of DHX9 as compared to control, but the reciprocal IP in Figure 4D showed no statistical significance between control and RNase H treatment. In Figure 4D, we hypothesize that NRL may account for only a small proportion of DHX9’s interactome, so the change in NRL levels could not be detected due to the sensitivity of our assay. DHX9 likely constitutes a large proportion of NRL’s interactome in HEK293 cells, hence the change in DHX9 level was more obvious when pulling down with NRL. We added this information to the results. See page 8.

(5) The only data supporting the claim that NRL interacts with R-loops are presented in Figure 5A.

Additional evidence that NRL interacts with R-loops comes from DRIP-Seq experiments where signals from R-loops overlap with NRL ChIP-Seq signals (Figure 7A). This shows that R-loops and NRL co-occur on multiple genomic regions. In addition, indirect evidence of NRL and R-loops’ interaction is shown in pull down experiments and PLA assays where R-loops influence DHX9 and NRL binding. We clarified this in the discussion. See page 14.

This is a co-IP of R-loops and then blotting for NRL, DHX9, and DDX5. Here, there is no signal for DDX5, quantification of DHX9 signal shows no statistically significant difference between RNase H treated and untreated samples, while NRL shows a signal in RNase H treated sample. These data are not sufficient to make the statement regarding the interaction of NRL with R-loops.

Thank you for this comment. We respectfully disagree as we observe statistically significant enrichment for both NRL and DHX9 in these experiments (See Fig5A). Some NRL continues to bind to DNA that is pulled down nonspecifically, which may be expected since NRL is a transcription factor. See for example R-loop binding by the transcription factor Sox2 (PMID: 32704541). However, binding to R-loops is evidenced by an enrichment compared to RNase H-treated sample. We clarified this in Results section (See page 9).

(6) Regarding R-loop mapping, the data analysis is quite confusing. The authors perform two different types of analyses: either overall narrow and broad peak analysis or strand-specific analysis. Given that the authors used ssDRIP-seq, which is a method designed to map R-loops strand specifically, it is confusing to perform different types of analyses.

Thank you for highlighting this point. This has enhanced the clarity of the methods and enriched the discussion. We aimed to identify R-loops as accurately as possible. We conducted two types of analyses to capture different aspects of R-loops: one that looks at overall patterns (narrow and broad peaks) and another that focuses on specific strands of DNA.

Using ssDRIP-seq, which is designed to map R-loops on specific strands, allowed us to examine R-loops formed in only one strand and those formed on both strands. To identify strand-specific R-loops, we filtered our RNase-H enriched peaks for those enriched on one strand compared to the opposite strand. We clarified the analysis in the results section, and Figure 6B. See page 10 and methods section page 25.

Next, the peak analysis is usually performed based on the RNase H treated R-loop mapping; what does it mean then to have a pool of "Not R-loops", see Figure 6B?

The “Not R-loop” group refers to peaks called using the opposite strand that are not observed when calling peaks using RNase H as control. We modified this figure for clarity (Figure 6B).

In that regard, what does the term "unstranded" R-loops mean? Based on the authors' definition, these are R-loops that do not fall within the group of strand-specific R-loops. The authors should explain the reasons behind these types of analyses and explain, what the biological relevance of these different types of R-loops is.

Thank you for helping us clarify this point. Unstranded R-loops are DNA regions containing DNA:RNA hybrids on both plus and minus strands and possibly representing bidirectional transcription by Pol II. We observed that unstranded R-loops are enriched only in intergenic regions, H3K9me3 regions, and downstream of the transcriptional termination site (TTS). We added to the discussion the possible implications of these enrichments, including regulation of Pol II termination and transcription of long genes. See Page 13.

(7) It would be more useful to show the percent distribution of R-loops over the different genomic regions, instead of showing p-value enrichment, see Figure 6C.

Since most of the genome is non-coding, plotting the distribution as a proportion was not informative since the vast proportion of the data falls in intergenic regions. However, we created a new figure showing observed vs. expected ratio that seems to be more informative and moved the current p-value figure to the supplement in revised version. See Figure 6C and S6D.

(8) Based on the model presented, NRL regulates R-loop biology via interaction with RBPs, such as DHX9, a known R-loop resolution helicase. Given that the gene targets of NRL TF are known, it would be useful to then analyze the R-loop mapping data across this gene set.

Thank you for this suggestion. We performed an analysis of R-loops on NRL-regulated genes. Interestingly, NRL target genes have an enrichment of stranded R-loops at the promoter/TSS and unstranded R-loops on the gene body compared to all Ensembl genes (Figure S7B). We added a table containing all NRL-regulated genes we used for this analysis (table S5) and a figure showing this result (Fig. S7B).

**Reviewer #2 (Public review):**
Summary:The authors utilize biochemical approaches to determine and validate NRL protein-protein interactions to further understand the mechanisms by which the NRL transcription factor controls rod photoreceptor gene regulatory networks. Observations that NRL displays numerous protein-protein interactions with RNA-binding proteins, many of which are involved in R-loop biology, led the authors to investigate the role of RNA and R-loops in mediating protein-protein interactions and profile the co-localization of R-loops with NRL genomic occupancy.Strengths:Overall, the manuscript is very well written, providing succinct explanations of the observed results and potential implications. Additionally, the authors use multiple orthogonal techniques and tissue samples to reproduce and validate that NRL interacts with DHX9 and DDX5. Experiments also utilize specific assays to understand the influence of RNA and R-loops on protein-protein interactions. The authors also use state-of-the-art techniques to profile R-loop localization within the retina and integrate multiple previously established datasets to correlate R-loop presence with transcription factor binding and chromatin marks in an attempt to understand the significance of R-loops in the retina.Weaknesses:In general, the authors provide superficial interpretations of the data that fit a narrative but fail to provide alternative explanations or address caveats of the results. Specifically, many bands are present in interaction studies either in control lanes (GST controls) of Westerns or large amounts of background in PLA experiments.

We have added additional information to the text regarding the presence of background signals in pull downs. We wish to note that experimental samples always exceeded background signals. We believe that reporting these raw findings (rather than showing shorter exposures) is valuable for the scientific community. We did not observe any background in the proximity ligation assay (PLA) that exceeded what is typically expected, and the signals were clearly discernible. Cases where signals are weaker, such as with DDX5, have been highlighted. In addition, we added a DHX9-IgG negative control for the human PLA experiment. See page 5 and Fig. S2B.

Additionally, the lack of experiments testing the functional significance of Nrl interactions or R-loops within the developing retina fails to provide novel biological insights into the regulation of gene regulatory networks other than, 'This could be a potentially important new mechanism'.

We agree that functional experiments are necessary to understand the molecular mechanisms behind R-loop regulation in the retina; however, we believe it goes beyond the scope of this initial characterization (as this is the first report on R-loops in the retina). We are currently pursuing these studies.

We performed new analysis on NRL-regulated genes as suggested by reviewer 1. We show that NRL target genes have an enrichment of stranded R-loops at the promoter/TSS and unstranded R-loops on the gene body compared to all Ensembl genes (Figure S7B), providing further evidence of the functional interaction between NRL and R-loops. See table S5 and Fig. S7B, and discussion.

Additionally, the authors test the necessity of RNA for NRL/DHX9 interactions but don't show RNA binding of NRL or DHX9 or the sufficiency of RNA to interfere/mediate protein-protein interactions. Recent work has highlighted the prevalence of RNA binding by transcription factors through Arginine Rich Motifs that are located near the DNA binding domains of transcription factors.

We agree that the role of RNA in these complexes is very exciting, and we are currently pursuing these studies. However, we believe that they fall outside the scope of this initial report on R-loops in the retina.

**Recommendations for the authors:**

**Reviewer #1 (Recommendations for the authors):**
There are a couple of minor comments:(1) Unfinished sentence; page 11, the end of the first paragraph.

Thank you for catching this error. We removed the unfinished text.

(2) Page 6: Figure S2A should be Figure S2.In general, the manuscript would benefit from a deeper explanation of the biological relevance of R-loop formation and the connection to NRL TF and the expression of genes regulated by NRL. In this regard, a more substantial description of the model would be useful.

We have modified the discussion for clarity and included new ideas on possible roles of R-loops in gene regulation of photoreceptors.

**Reviewer #2 (Recommendations for the authors):**
(1) The specificity of interactions needs to be addressed:- Figure 1B - HNRNPUI1 bands present in GST control.- Figure 1C - Bands present in the Empty Vector control IP for HNRNPU and DHX9.- Supplemental Figure 1A - most proteins are present in GST control suggesting prevalent binding to GST and lack of specificity for other interactions.

Thank you for your comment. RNA-binding proteins can have more background as observed in other studies (e.g. PMID: 32704541) but there is always a higher signal in experimental samples compared to controls. While we agree that we can enhance the conditions for immunoprecipitation (IP) by optimizing washing buffers, exposure and other parameters, we believe the current methods tell the story. We have added additional text explaining this. See page 5.

(2) Use of the term 'Strongest' interaction - IPs don't directly address the strength of interaction, but depend on levels of expression AND affinity. The strength of interaction should be tested using techniques like an OCTET or SPR assay. One can also quantify the effect that RNA would have in such an assay.

Thank you for your suggestion. We replaced the term 'stronger' with “higher signal” and “robust” at most places. The source of protein lysates is the same for experiments and controls, thus the amount of protein is consistent in both conditions, and not dependent on level of gene expression.

(3) In supplemental tables, please use the proper gene names, not the UniProt peptide name. For example, there are no genes named ELAV1-ELAV4. These should be ELAVL1-ELAVL4. A short glance identifies >10 gene name errors.

Thank you for the suggestion. We updated current gene names in all tables.

(4) Please provide the rationale for the choice of DNA sequence for the DHX9 nucleotide sequence used for EMSA assays. In the human DHX9 locus, the NRL ChIP-seq peak looks to be contained in Intron1 whereas the NRL ChIP-seq peak in mouse DHX9 looks to be in the proximal upstream promoter. Did the authors choose an evolutionarily conserved sequence in the promoter region that contained the NRL motif or does the probe sequence arise from the sequence that has known NRL binding as assayed by NRL ChIP-seq? A zoomed-in image of the NRL ChIP-seq pile-ups in the DHX9 locus in each species would be beneficial.

Thank you for this suggestion. The probe was chosen by scanning for NRL binding motifs on the Chip-Seq peak at the human DHX9 promoter. We added a Zoom-in image of the ChIP-Seq or CUT&RUN reads for NRL on both human and mouse retinas. Figure 3D shows NRL binding in both species in regions containing the homologous motif. The sequence is partially conserved and shown in the figure.

(5) Normalization in RNaseH/RNaseA Co-IP experiments. Why does RNAseH treatment result in increased NRL IP (increased NRL expression?) or does RNaseA treatment cause reduced IP of DHX9? These differences seem to cause a 'denominator' effect, leading the Authors to conclude decreased co-IP of DHX9 with NRL when R-loops are inhibited or increased co-IP of NRL with DHX9 when RNA is degraded. An alternate interpretation would be that inhibiting the R-loop binding of NRL unmasks the epitope for antibody recognition. The authors should test NRL binding to RNA and determine if RNA binding affects the co-IP of NRL with DHX9.

We agree that removing total RNA by RNase A or R-loops by RNase H may alter the accessibility of our antibodies to the epitopes, resulting in the differences in the level of total protein pulled down. However, we quantified the relative level of the associating protein to the total protein and confirmed, in reciprocal assays, that RNase A treatment led to increased interaction between NRL and DHX9. However, the quantification was not consistent between the reciprocal IPs upon RNase H treatment. We reason that in Figure 4D, as NRL may account for only a small proportion of DHX9’s interactome, the change in NRL level could not be detected due to the sensitivity of our assay. However reciprocally, DHX9 can constitute a larger proportion of NRL’s interactome in HEK293 cells, hence the change in DHX9 level was more obvious. We added this information to the text. See page 8.

(6) Figure 7 - Malat1 - there doesn't seem to be an overlap of NRL with Stranded R-loop peaks in this image. Nrl seems to flank the region of R-loops.

We changed Malat1 for Mplkip that shows a direct overlap of Nrl binding and R-loops. See Figure 7C.

(7) Results end with 'A Model'. Seems like some concluding remarks and references to Figure 8 were mistakenly left out.

Thank you for catching this typo. We removed the misplaced text.

(8) Model and Discussion - authors should show raw data for RHO with respect to NRL binding and R-loops. No evidence was provided regarding R-loops (or lack thereof) in the Rhodopsin locus. Additionally, conclusions stating that "R-loops... are specifically depleted from genes, such as Rhodopsin, with high expression levels" go against Figures 7B and 7C. Malat1 is one of the highest expressed genes in the retina and contains R-loops.

Thank you for helping us clarify our hypothesis. We added a genome browser view of Rhodopsin showing the absence of R-loops (Fig. S8). We hypothesize that R-loops could interfere with achieving higher rates of transcription, however we did not mean to say that all high expressed genes lack R-loops. We have rephrased the discussion to clarify this point.

(9) Neuronal genes, particularly those involved in synaptic transmission are known to be, on average, longer than most genes (Gabel, 2015; PMID: 25762136). Is it possible that R-loops are detected at genes involved in synaptic function/structure solely because of transcript length, as it takes longer for transcription termination to resolve in genes that are longer? A plot showing R-loop enrichment and transcript length would address this.

We added a plot showing gene length in relation to R-loops and expression levels. We observed that R-loops are more common over long genes regardless of their expression levels. We also observed that the concomitant presence of stranded and unstranded R-loops is restricted to the longest genes in most cases. We added this to Figure 7D.